# CO₂ and summer insolation as drivers for the Mid-Pleistocene transition

Meike D. W. Scherrenberg[1], Constantijn J. Berends[1], Roderik S.W. van de Wal[1,2]

[1]Institute for Marine and Atmospheric research Utrecht, Utrecht University, Utrecht, the Netherlands
[2]Faculty of Geosciences, Department of Physical Geography, Utrecht University, Utrecht, the Netherlands

*Correspondence to*: M.D.W. Scherrenberg (M.D.W.Scherrenberg@uu.nl)

During the Mid-Pleistocene transition (MPT; ~1.2–0.8 million years ago) the dominant periodicity of glacial cycles increased from 41 thousand years (kyr) to an average of 100 kyr, without any appreciable change in the orbital pacing. As the MPT is

not a linear response to orbital forcing, it must have resulted from feedback processes in the Earth system. However, the precise mechanisms underlying the transition are still under debate.

In this study, we investigate the MPT by simulating the Northern Hemisphere ice sheet evolution over the past 1.5 million years. The transient climate forcing of the ice-sheet model was obtained using a matrix method, by interpolating between two snapshots of global climate model simulations. Changes in climate forcing are caused by variations in $CO_2$,

insolation, as well as implicit climate–ice sheet feedbacks.

Using this method, we were able to capture glacial-interglacial periodicity during the past 1.5 million years and thereby reproduce the shift from 41 kyr to 100 kyr cycles without any additional drivers. Instead, the modelled frequency change results from the prescribed $CO_2$ combined with orbital forcing, and ice sheet feedbacks. Early Pleistocene terminations are initiated by insolation maxima. After the MPT, low interstadial $CO_2$ levels may compensate insolation maxima which

would otherwise favour deglaciation, leading to a longer duration of the glacial cycle. Terminations are also affected by ice volume. If the North American ice sheet is small or very large, it becomes sensitive to small temperature increases. A medium sized ice sheet is less sensitive through its location and the merger of the Laurentide and Cordilleran ice sheets. Therefore, Late Pleistocene terminations are also facilitated by the large ice-sheet volume, were small changes in temperature lead to self-sustained melt.

Additionally, we carried out experiments with constant $CO_2$, where we can capture the 41-kyr cycles and some Late Pleistocene cycles. However, no persistent 100-kyr periodicity is established. Experiments with constant (or evolving) $CO_2$ concentrations did not generate a substantial precession signal in the ice volume. Instead, the frequency is dominated by successful terminations, which are initiated by strong (generally obliquity) insolation maxima. Our results therefore indicate that the glacial cycle periodicity of the past 1.5 million years can be described by changes in insolation, $CO_2$ and ice sheets

feedback processes, and that maintaining low $CO_2$ throughout insolation maxima may prolong glacial cycles.

## 1. Introduction

During the MPT, the duration of glacial cycles shifted from 41 kyr to an average of 100 kyr, but the main mechanisms behind this change are still under debate. Glacial cycles are paced by orbital cycles, namely precession (~19/23 kyr), obliquity (~41 kyr), and eccentricity (~98/400 kyr), which determine the latitudinal and seasonal distribution of solar radiation received by the Earth. Ice sheets are especially sensitive to summer insolation, as regions may undergo melt and a small change in temperature or insolation can strongly alter melt rates. As a result, the 41 kyr periodicity during the Early Pleistocene (2580 – 800 kyr ago) mostly follows the obliquity cycle (e.g., Huybers and Tziperman, 2008; Tabor et al., 2015; Watanabe et al., 2023), but the mechanisms behind the average 100 kyr periodicity of the Late Pleistocene (800 – 11 kyr ago) are more difficult to explain. It has been suggested that the 100 kyr periodicity is a non-linear response to predominantly obliquity (e.g., Huybers and Wunsch, 2005), precession and eccentricity (e.g., Lisiecki, 2010; Hobart, et al. 2023; Blackburn et al., 2024), or a combination of orbital cycles (e.g., Huybers, 2011; Feng and Bailer-Jones, 2015; Tzedakis, et al. 2017). Nevertheless, the transition from 41 kyr to 100 kyr glacial cycles took place without any considerable change in the orbital cycles, and therefore feedback processes within the Earth's system must have contributed to the MPT.

One overarching hypothesis that could partially explain the MPT concerns ice-sheet threshold regimes (see Berends, et al. 2021a; Paillard, 1998): Small or flat ice sheets can easily melt at insolation maxima. Medium-sized ice sheets may survive insolation maxima due to albedo and topography feedbacks, facilitating low temperatures in glaciated regions. Large ice sheets become vulnerable through positive feedbacks such as the elevation-temperature feedback, albedo feedbacks, and high basal temperatures enhancing sliding (Bintanja and van de Wal, 2008), and Proglacial Ice Sheet Instability (PLISI; see Quiquet et al., 2021; Hinck et al., 2022; Scherrenberg et al., 2024). This is further supported by several studies showing that the Late Pleistocene glacial cycles only melt once they reach a certain ice volume (Parrenin and Paillard, 2003; Bintanja and van de Wal, 2008; Abe-Ouchi et al., 2013; Verbitsky et al., 2018; Berends et al., 2021a). These threshold regimes can therefore act as a precondition that facilitates the MPT, but require another process (e.g., long term cooling) to prompt a shift in the ice volume rhythm.

It has also been suggested that the MPT is caused by regolith removal, as first proposed by Clark and Pollard (1998). The regolith hypothesis states that sediments covered North America during the Early Pleistocene. Sediments are easily deformed and enhance sliding, creating flatter ice sheets with relatively larger ablation areas. Therefore, if ice sheets are superimposed on sediments, they are more vulnerable to small changes in forcing. Once this sediment was removed by the erosive action of the ice sheet, friction increased, reducing ice sheet flow. This produces thicker ice sheets that may survive insolation maxima. Several modelling studies were dedicated to this regolith hypothesis and were able to capture characteristics of the MPT (e.g., Tabor and Poulsen 2016, Ganopolski and Calov, 2011; Mitsui et al., 2023). Recently, Willeit et al. (2019) used a coupled climate-ice-sheet-carbon-cycle model and was able to reproduce the MPT using prescribed gradual atmospheric $CO_2$ decrease and regolith removal. The regolith theory is also supported by geological data which show a change in the composition of glacial sediments (e.g., Roy et al., 2004; Portier et al., 2021).

Alternative explanations argue that the carbon cycle plays a major role in controlling the transition from 41 kyr to 100 kyr glacial cycles. During the Late Pleistocene glacial cycles, glacial-interglacial variations in $CO_2$ were roughly 90 ppm. (Bereiter et al., 2015). This variation in $CO_2$ can be largely attributed to the ocean (e.g., Sigman and Boyle, 2000; Brovkin et al., 2012). During glacial periods, $CO_2$ solubility increases due to lower ocean temperatures, which is further enhanced by increased alkalinity (Kurahashi-Nakamura et al., 2010; Sigman et al., 2010). Glacial-erosional and enhanced dust concentrations in the atmosphere provides nutrients to the Southern Ocean. This increases the biological productivity (Martin, 1990; Martínez-García et al. 2014; Chalk et al., 2017; Saini et al., 2023) and eventually leads to more uptake of $CO_2$ in the deep-ocean. Additionally, decreased deep-ocean ventilation during glacial periods may have more efficiently trapped carbon (Hasenfratz et al., 2019), which could be explained by increased sea ice extent (Menviel, 2019), or enhanced ocean stratification (Bouttes et al., 2009; Adkins, 2013; Qin et al., 2022). These processes in the carbon cycle are important for decreasing $CO_2$ levels in the atmosphere during Late Pleistocene glacial periods, but perhaps they also played a crucial role during the MPT. After the MPT, glacial $CO_2$ concentrations may have dropped due to increased deep-ocean carbon storage (Köhler and Bintanja, 2008; Lear et al., 2016; Farmer et al., 2019; Qin et al., 2022; Thomas et al., 2022) which could, for example, have resulted from increased Antarctic bottom water formation due to a change in circulation (e.g., Pena et al., 2014), or an increase in sea ice extent (Detlef et al., 2018).

Nevertheless, despite the complexity and uncertainties in the carbon cycle, atmospheric $CO_2$ can be measured from ice cores (e.g. Bereiter et al. 2015), or estimated through proxies (e.g. Hönisch et al., 2009; Da et al., 2019; Dyez et al., 2018; Yamamoto et al., 2022). Ice core $CO_2$ records tend to have low uncertainties, as $CO_2$ can be measured directly from bubbles or clathrates in the ice, which contain a snapshot of the paleo-atmosphere. However, the current oldest continuous record dates back to only 800 kyr ago (Bereiter et al., 2015) which does not capture the MPT. $CO_2$ can also be estimated indirectly from certain isotope ratios (e.g. boron isotopes; see Hönisch et al., 2009; Dyez et al., 2018; Da et al., 2019; or leaf-wax; Yamamoto et al., 2022), allowing for $CO_2$ records that extent beyond the ice-records. However, to generate a $CO_2$ record from a proxy requires physical and chemical assumptions, which results in a higher uncertainty. Recently, Yamamoto et al. (2022) published a reconstruction based on a leaf-wax indicator, which was calibrated to the 800 kyr ice core record. The record by Yamamoto et al. (2022) is currently the only continuous $CO_2$ proxy that dates back to almost 1.5 million years ago, and thus provides a new opportunity for modelling studies on the MPT.

In this study, we simulate the past 1.5 million years using an ice-sheet model without any change in model set-up over time, and a constant sediment mask. Our main goal is to explore if we can simulate the frequency change during the MPT, and the possible mechanisms behind it, based on only prescribed $CO_2$ and insolation variations. The purpose is therefore not to make perfect spatial and size reconstructions, but rather to explain the frequency change of glacial-interglacial variability. To provide the model with information on climate, we use a matrix method which is based on interpolated 2D-time-slices from general circulation models (GCM). The temporal climate interpolation is driven by prescribed $CO_2$ and insolation (see Berends et al., 2018; Scherrenberg et al., 2024). Here we use for the first time the long $CO_2$ record from Yamamoto et al. (2022), which covers the MPT, in ice-sheet model simulations. This method allows us to provide transient climate forcing at a significantly

reduced computational time compared to GCM's or intermediate complexity models (e.g., Ganopolski and Calov, 2011; Willeit et al., 2019). Temperature change is mainly driven by prescribed $CO_2$ records from leaf-wax proxy (1450-0 kyr ago)
combined with caloric summer half-year insolation (Tzedakis et al. 2017). To establish the importance of $CO_2$ and insolation variations on glacial-interglacial time-scales, we run the same 1450 kyr experiments with constant $CO_2$ or insolation levels.

## 2. Methods

To simulate ice-sheet evolution during the past 1.5 million years, we use the vertically integrated ice-sheet model IMAU-ICE version 2.1 (Berends et al., 2022). North America, Eurasia and Greenland are simulated in separate model domains, which are
shown in Fig. 1.

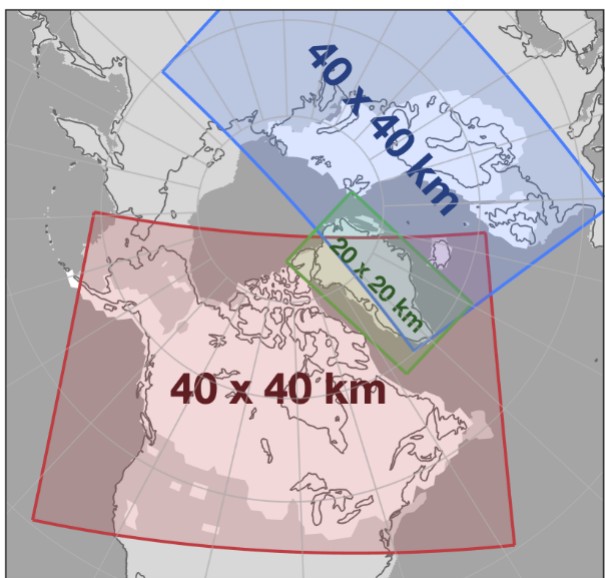

**Figure 1.** The extent and resolution of the ice-sheet model domains: North America (red), Greenland (green) and Eurasia (blue). For reference, the ICE-6G LGM ice sheet of Peltier et al. (2015) is shown in white. Ice in overlapping regions is removed (e.g., Greenland in the North American domain).

The flow of ice is calculated using the shallow ice / shallow shelf approximation (Bueler and Brown, 2009). The sliding of ice is calculated using the Budd-type sliding law by Bueler and van Pelt (2015). Basal hydrology is based on Martin et al. (2011). To calculate basal friction, we apply a present-day sediment map for North America (Gowan et al., 2019), and, as this map does not cover Eurasia, we generate a friction map for this continent using the sediment thickness by Laske and Masters (1997). Eurasian regions with less than 20 m sediment thickness receive a till friction angle of 10°, while other regions
receive a 30° till friction angle. These friction maps a static, but a simulation with a homogenous sediment coverage is presented in the supplementary material. To simulate bedrock changes due to ice sheet load, we use an Elastic Lithosphere, Relaxing Asthenosphere model (Le Meur and Huybrechts, 1996). If bedrock topography is below sea level, it is considered as

ocean or lake. At the grounding line we include a sub-grid friction scaling scheme based on Leguy et al. (2021) and Feldmann et al. (2014). As such, a small change in grounding line position can prompt a significant change in friction, capturing Marine and Proglacial Lake Ice Sheet Instability. Berends et al. (2022) has shown that this scheme is capable of resolving the sub-grid grounding-line position, and produce the grounding-line positions and retreat rates that are within the model ensemble for the MISMIP+ intercomparison experiment (Conford et al. 2020). Ice is removed if the thickness at the calving front is below 200 m. Additionally, floating ice beyond the continental shelf is always removed. To calculate basal melt, we use a depth dependent sub-shelf parameterization based on Martin et al. (2011). Ocean temperatures are based on de Boer et al. (2013), where we interpolate spatially homogeneous ocean temperatures based on $CO_2$ and insolation. The ocean temperatures therefore evolve over time, but they are homogeneous within the model domains. For the surface mass balance (SMB), we use IMAU-ITM (insolation, temperature model), which includes a melt parameterization (Bintanja et al., 2002) that depends on temperature, albedo and insolation, a temperature-based snow-rain partitioning scheme (Ohmura et al., 1999), and a refreezing scheme (Huybrechts and de Wolde, 1999). The latter is calculated by limiting the amount of refreezing to the available liquid water, temperature and firn depth. IMAU-ITM has been shown to be perform well for present-day Greenland conditions (see Fettweis et al., 2020).

## 2.1 Climate forcing

To calculate the transient climate forcing over the past 1.5 million years, we interpolate between GCM-calculated climates of pre-industrial (PI) and the Last Glacial Maximum (LGM). Since the choice of GCM climates can cause large differences in the modelled ice sheets (Alder and Hostetler, 2019; Niu et al., 2019; Scherrenberg et al., 2023), we use a climate ensemble obtained from the Paleoclimate Modelling Intercomparison Project (PMIP4; Kageyama et al., 2017, 2018) rather than a large quantity of time-slices from a single model. The four simulations that had all data necessary for our simulations are MIROC (Ohgaito et al., 2021), MPI (Mauritsen et al., 2019), AWI (Shi et al., 2023) and INM (Volodin et al., 2018). Differences in topography between climate and ice-sheet model are accounted for by applying a lapse rate correction for temperature, and by applying a correction for precipitation with wind-ward and leeward topography effects based on the approach by Roe and Lindzen (2001). The technical details of these methods are described in Scherrenberg et al. (2023).

To interpolate the climate time-slices through time we use a matrix method (see Berends et al., 2018; Scherrenberg et al., 2023; Scherrenberg et al., 2024). Equations governing the matrix method are described in Appendix A. Here we provide a brief, qualitative summary.

Temperature depends on the prescribed external forcing ($CO_2$ and summer insolation) and the modelled ice-sheet, to create a first-order approximation of the ice-albedo feedback. Fig. 2 shows how the prescribed $CO_2$ and caloric summer insolation (Tzedakis et al. 2017) are combined to calculate the external forcing index. To approximate the albedo feedback, we follow the approach by Berends et al. (2018), where the monthly/latitudinally varying insolation is multiplied with the modelled albedo to yield the 'absorbed insolation'. As the modelled albedo depends on the modelled ice-sheet extent and snow cover, this introduces the feedback from the ice sheet back onto the climate.

Precipitation is modelled similarly, but there the ice-sheet term relates to the (modelled, local) ice thickness rather than the extent, in order to approximate the plateau desert effect. A geometry-based correction, which includes the orographic forcing of precipitation by upslope winds (Roe and Lindzen, 2001), helps to more accurately track the higher precipitation rates at the (moving) ice margin.

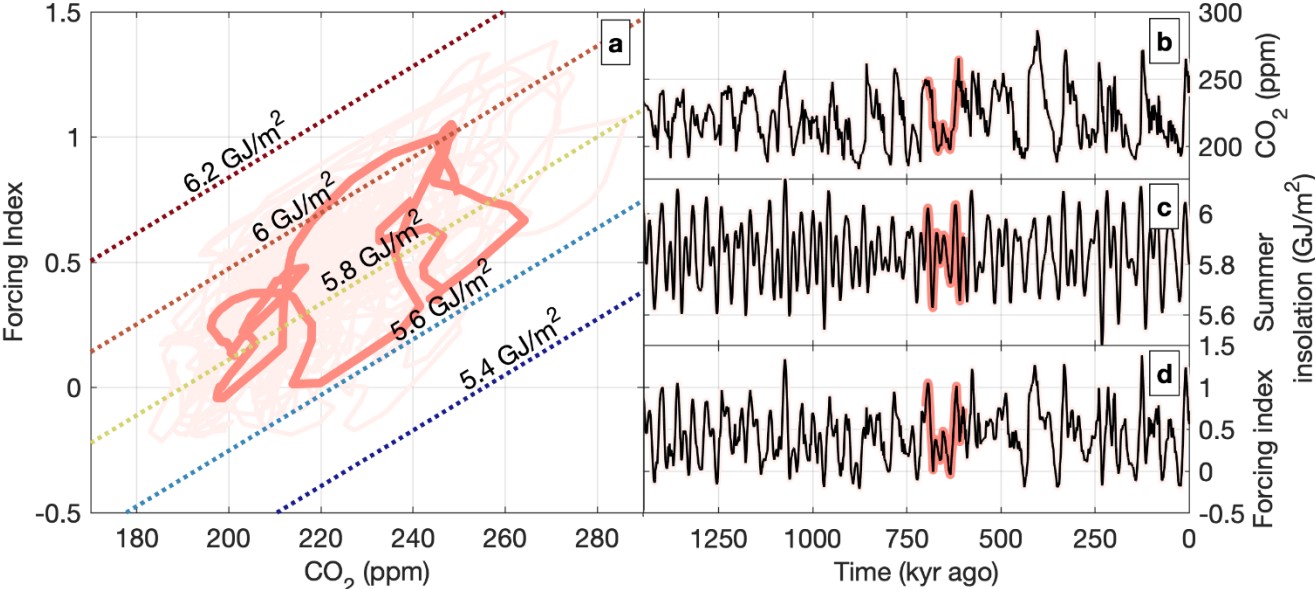

**Figure 2.** The forcing index (a,d) is a combination of $CO_2$ (b; x-axis in a) and caloric summer-half year insolation at 65°N (c; diagonal lines in a). Red shows the evolution of the forcing index over one glacial cycle, while pink (see panel a) shows the forcing index over the entire simulation.

**2.2 Benthic δ¹⁸O**

Challenges in studying the MPT are the large uncertainties in sea level reconstructions, especially during the Early Pleistocene. The benthic $\delta^{18}O$ record contains the combined signals of ice volume and deep-water temperature, which are difficult to disentangle. To validate our results, we simulate benthic $\delta^{18}O$ by modelling a separate contribution from deep-water temperature and ice volume.

IMAU-ICE includes a $\delta^{18}O$ model based on the approach by de Boer et al. (2013). This approach uses a depth-integrated advection solver to calculate the evolution of the englacial isotope content, which is forced at the surface by an elevation-dependent parameterisation following Clarke et al. (2005). The governing equations of this approach are detailed in appendix B. The benthic $\delta^{18}O$ contribution from ice volume is calculated by integrating the modelled englacial isotope content over all three modelled ice sheets.

To obtain the deep-water temperature we calculate global temperatures based on $CO_2$, and we apply a 3000-year running-mean to reflect the lag between the atmosphere and deep-ocean. We then linearly convert deep-water temperatures to deep-water $\delta^{18}O$ contribution.

## Results

In this section, we show the results from our 1.5-million-year simulations. First, we present our baseline simulation, after 175 which we explore the mechanisms behind the simulated MPT. In the last section we present experiments with either constant insolation or constant $CO_2$.

### 3.1 Baseline results

We conduct the 1.5-million-year baseline simulation, where temperature evolves with prescribed $CO_2$ from leaf-wax proxy (Yamamoto et al., 2022), caloric summer insolation (Tzedakis et al., 2017), and an albedo feedback. In Fig. S1, we show 180 results from a 0.8-million-year simulation which is forced by the ice core $CO_2$ record from Bereiter et al. (2015) instead. The results of that simulation are very close to those of the Baseline simulation, as the two $CO_2$ records are very similar (reflecting the fact that the leaf-wax based reconstruction was calibrated to the ice core record).

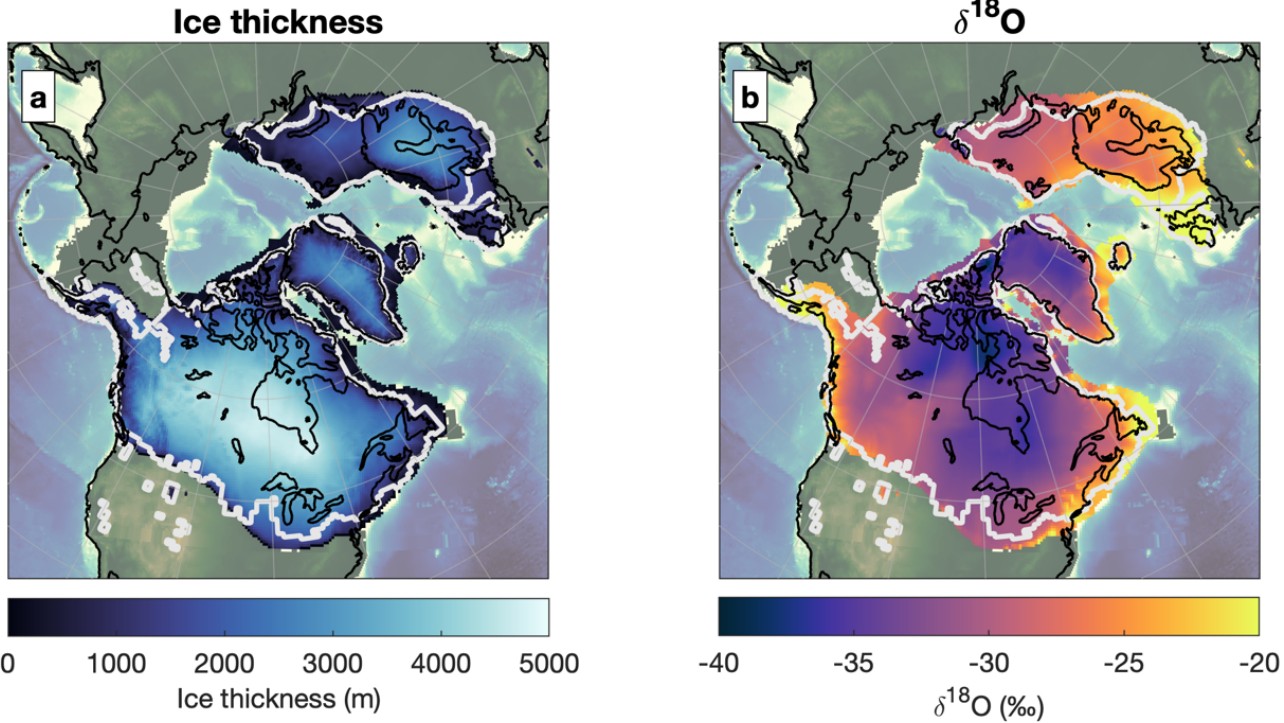

**Figure 3.** The modelled ice thickness (a), and englacial $\delta^{18}O$ (b) at 20 kyr ago in the baseline simulation. The reconstruction of the extent 185 by Peltier et al. (2015) is shown as white contours. The present-day coastline is shown in black.

Fig. 3 shows that the LGM ice extent reasonably matches the ICE6G ice volume reconstruction by Peltier et al. (2015; see white contours). The englacial $\delta^{18}O$ (Fig. 3b) is less depleted where temperatures are high and elevation is low, such as the margins. Fig. 4 shows time-series of the prescribed summer insolation (a), $CO_2$ (b), the modelled sea level change (c), and benthic $\delta^{18}O$ (d), which are compared to reconstructions by Spratt and Lisiecki et al. (2016), Rohling et al. (2021) and Ahn et al. (2017). In this figure, the modelled sea-level change is increased by 20% to reflect processes that were not explicitly modelled, such as ice volume changes in Antarctica, and temperature/density changes of the ocean. This is justified by the strong correlation between global climate, Northern Hemisphere ice volume, and Antarctic ice volume (Gomez et al., 2020), and does not account for out-of-phase behaviour between the Northern and Southern Hemisphere. Additionally, low benthic $\delta^{18}O$ and high sea-levels cannot be simulated as we do not include Antarctica.

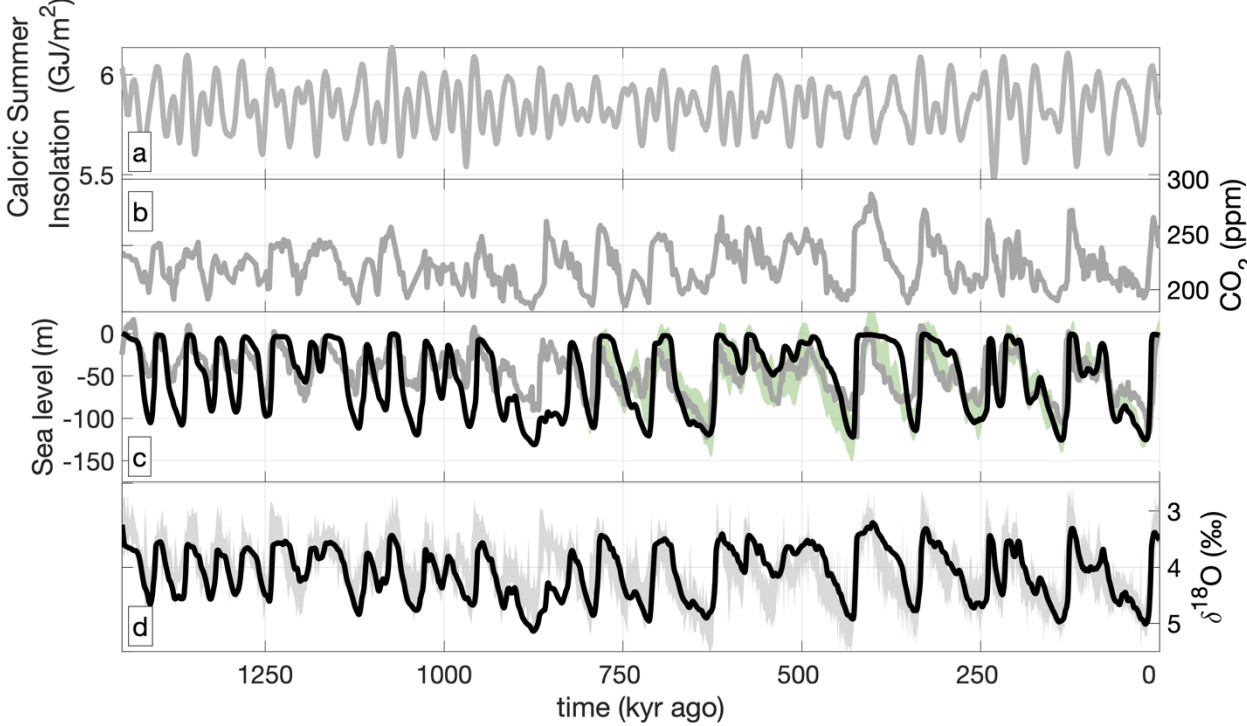

**Figure 4.** Time-series of the past 1.5 million years ago showing prescribed caloric summer half-year insolation (a) prescribed $CO_2$ forcing (b), sea level (c) and benthic $\delta^{18}O$ (d). Black lines show modelled sea level (c) and benthic $\delta^{18}O$ (d) of the baseline simulation. Gray shows the reconstructions of caloric summer half-year insolation by Tzedakis et al., (2017; a), $CO_2$ by Yamamoto et al., (2022; b) sea level by Rohling et al. (2021; c), and $\delta^{18}O$ by Ahn et al. (2017; d). In green (c), sea level reconstruction by Spratt and Lisiecki et al. (2016). Note that lines in grey and green represent reconstructions, while black represents model output.

The baseline simulation captures the $\delta^{18}O$ variability of most glacial cycles except for the termination at ~865 kyr (MIS 21), where we simulate partial retreat of the North American ice sheet instead of a full deglaciation. $CO_2$ is low during

the 866 kyr ago insolation peak, but rises while insolation strength decreases. Therefore, insolation compensates the rise in
$CO_2$, preventing a deglaciation.

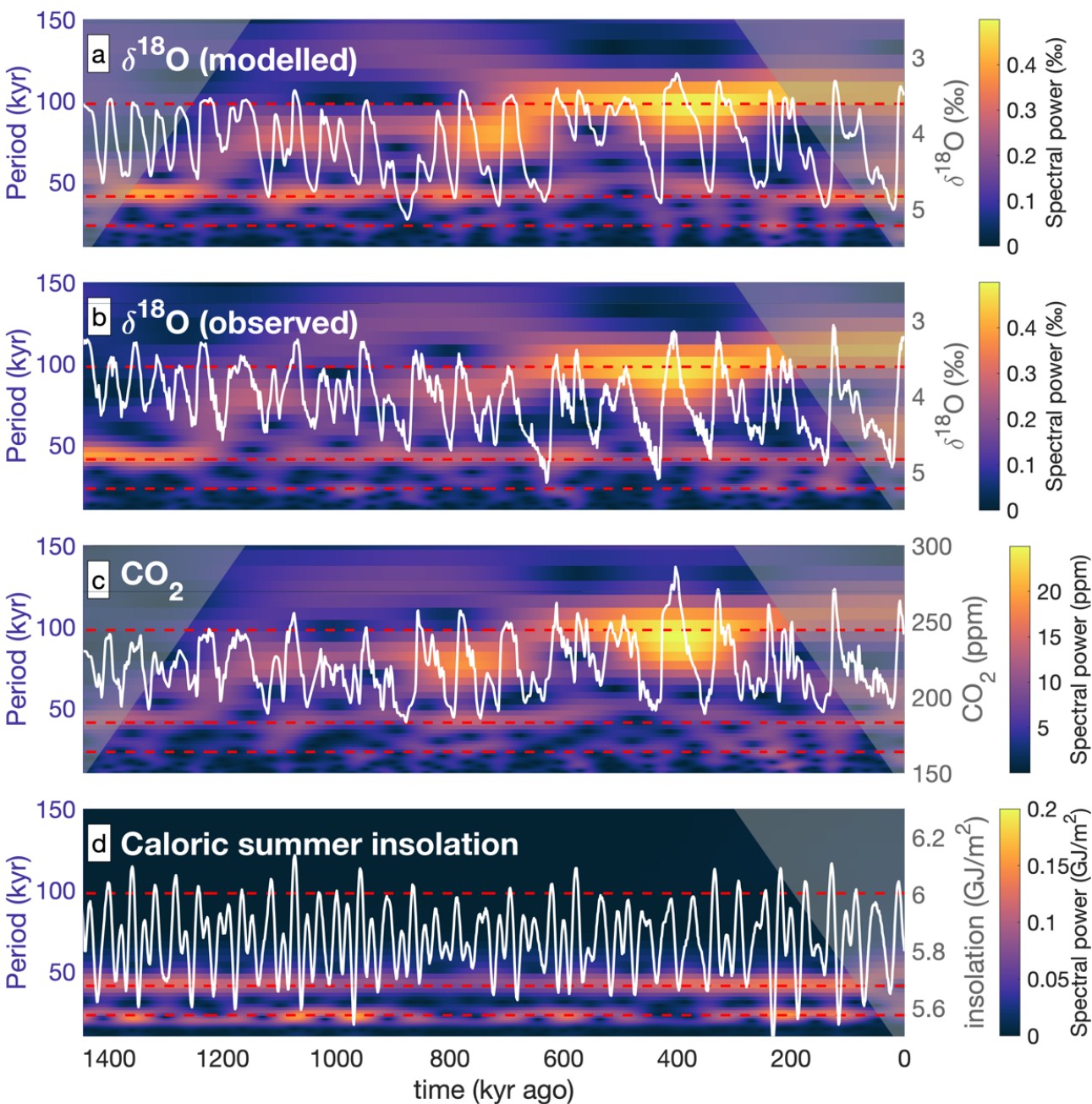

**Figure 5.** Wavelets transforms showing the explained variance and frequency of the modelled $\delta^{18}O$ (a), observed $\delta^{18}O$ (Ahn et al., 2017; b), $CO_2$ (Yamamoto et al., 2022; c) and caloric summer half-year insolation (Tzedakis et al., 2017; d) of the past 1.5 million years. Corresponding time-series are shown in white (see right y-axis). Red dotted horizontal lines indicate 98 kyr, 41 kyr and 23 kyr periodicities, representing
eccentricity, obliquity and precession.

Our baseline simulation captures the Late Pleistocene ice volume amplitude, but does not simulate a substantial change in amplitude during the past 1.5 million years. Average glacial peak Northern Hemisphere ice volume during 1,500-1,000 kyr ago is only slightly smaller (79±15 m.s.l.e; m sea level equivalent) compared to the 500-0 kyr ago (98±7 m.s.l.e.).

At the same time, Early Pleistocene $\delta^{18}O$ and sea levels at glacial maxima are often higher compared to the reconstructions by Ahn et al. (2017) and Rohling et al. (2021). To test if this large Early Pleistocene amplitude results from the relatively high present-day basal friction, we conduct a simulation with reduced basal friction (see Fig. S2). In this simulation we apply a homogenous "sediment" friction during the Early Pleistocene (until 800 kyr ago), and use the baseline's friction map for the Late Pleistocene. This sediment coverage is simulated by applying a 10° till friction angle to the North American and Eurasian

domains, which is equivalent to the sediment-covered regions in the baseline friction map. This simulation has a similar glacial-interglacial periodicity as the baseline, but shows a ~10% reduction in amplitude in the Early Pleistocene relative to the baseline. The reduced friction also makes the ice sheet more prone to collapse resulting in full deglaciation at 865 kyr ago, while the baseline does not.

As our baseline simulations captures the main glacial-interglacial periodicity during the past 1.5 million years, we

can explore the mechanisms behind the MPT in more detail.

**3.2 Mechanisms behind the Mid-Pleistocene Transition**

In the previous paragraph we showed that we can capture the glacial-interglacial periodicity in the baseline simulation. In this section we explore the key elements of the frequency change in our baseline simulation.

Fig. 5a shows a wavelet transform of our modelled $\delta^{18}O$ and compares it to the observed record (Fig. 5b). Our

simulation generates the transition from 41-kyr to 100-kyr glacial cycles, though with only a small change in glacial-interglacial ice volume difference. The periodicity in the baseline simulation results from the prescribed insolation and $CO_2$ forcing, combined with ice sheet feedbacks. Since the orbital cycles alone cannot explain all characteristics of the MPT (see Legrain et al., 2023), the main culprit behind our modelled MPT must be the prescribed $CO_2$ forcing combined with ice-sheet feedback processes.

We can compare the frequency spectrum in the modelled benthic $\delta^{18}O$ to the prescribed caloric summer insolation and $CO_2$ that drive the modelled temperature change. Fig 5c and 5d show wavelets of the caloric summer insolation and $CO_2$ forcing. The caloric summer insolation has a strong ~20 kyr and 41 kyr periodicity, corresponding to precession and obliquity respectively. The $CO_2$ forcing has a weak signal in the 41 kyr periodicity, but a strong 100 kyr signal in the Late Pleistocene. This change in frequency takes place when the modelled $\delta^{18}O$ also changes from 41 kyr to an average 100 kyr.

These results suggest a regime shift between the Early and Late Pleistocene. The 41 kyr periodicity of the Early Pleistocene is generated by the orbital cycles, as the ice sheet melts during strong summer insolation, which tends to correlate with obliquity maxima. The Late Pleistocene is more dominated by $CO_2$, though still paced by the orbital cycles. Terminations

take place if both $CO_2$ and insolation are high enough, while low interstadial $CO_2$ levels can sometimes partially compensate the strong summer insolation (e.g., 737, 175 and 50 kyr ago), leading to prolonged glacial periods.

Besides the climate forcing ($CO_2$ and insolation), the ice sheets themselves may also play a significant role in determining which insolation maxima lead to interstadials or terminations. Fig. 6a shows the ice volume of the North American ice sheet at the onset of all modelled terminations, as a function of the climate forcing (external forcing index) at that time. The termination onsets are defined as the maxima in the modelled ice volume that preceded an uninterrupted decrease to (near-) zero ice volume. These onsets therefore represent an integrated mass balance of around zero and a near-equilibrium with the

climate forcing, before the integrated mass balance becomes negative. This collection of terminations spans an S-shaped curve, indicating a non-linearity between ice sheet volume and climate forcing. If the ice sheet is small (<20 m.s.l.e.) it starts melting at relative warm climates. This changes above the 20 m.s.l.e. threshold and below the 60 m.s.l.e threshold. Within this regime, a small change in the climate forcing will substantially increase the ice volume threshold for glacial terminations. When exceeding the 60 m threshold, the S-curve gradually levels off again, and at ice volumes exceeding 75 m.s.l.e., any additional

cooling will barely increase the ice volume at terminations. Moreover, even under LGM-like conditions, these large ice sheets may start to lose mass, which could eventually lead to a deglaciation.

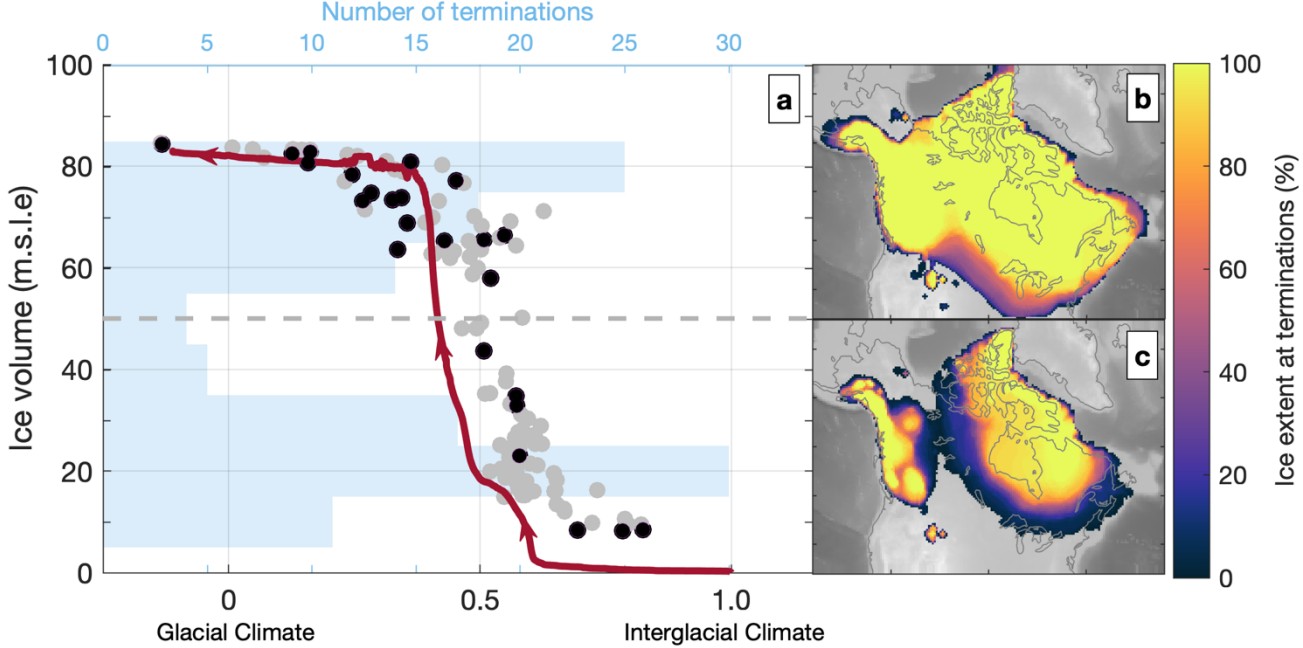

**Figure 6.** Panel a shows the climate forcing (forcing index: insolation and $CO_2$) and North American ice volume at the onset of deglaciations (a). Black circles represent the baseline simulation, while grey represents the simulations that are introduced in section 3.3 (constant $CO_2$ or

insolation). The red line belongs to the gradual cooling simulation. In blue, histograms showing the number of terminations per 10 m.s.l.e. ice volume bin. The extent of these terminations is shown in panel (b,c), either with ice volumes above (b) or below 50 m.sl.e. (c). The colours in b and c indicate which percentage of the terminations in these two groups were covered by ice when the mass balance became negative. We defined a deglaciation as a continuous melt phase leading to an ice volume of less than 8 m.s.l.e (~10 % of LGM ice volume).

The nonlinearity between ice volume and climate forcing is further explored by the gradual cooling simulation (see
the red line in Fig. 6a), where we gradually altered the climate from interglacial to glacial condition over a period of 1 million
years. This simulation provides the ice sheet enough time to readjust to any change in climate forcing, so that it is always
(nearly) in equilibrium. As the climate cools, first North-Eastern Canada and the Rocky Mountains will be covered by ice,
prompting the first increase in ice volume (<20 m). Afterwards, the majority of growth results from a (high latitude) west-
ward expansion of the Laurentide ice sheet, where it has space to grow driven by relatively cold climates. The growth of the
ice sheet is largely self-sustained by an increase in ice sheet height and albedo, and if allowed to fully adjust to the climate
forcing, creates the near vertical profile seen in Fig. 6a. Eventually, the Laurentide and Cordilleran merge together at an ice
volume of 54 m.s.l.e, leading to high growth rates due to the merging of ice flows and ablation areas. This merging may also
explain the relatively low number of terminations around 50 m.s.l.e., as indicated by the blue histogram in Fig. 6a. We can see
this in Fig. 6b and 6c, where the extent at the onset of every termination event below or above 50 m.s,l.e. are combined. This
50 m.s.l.e. boundary groups most of the separated or merged states of the Cordilerran and Laurentide ice sheets. After the
merging, the ice sheet cannot migrate west or north, but is forced to thicken or grow to the warmer south, which eventually
slows down the growth and prevents further ice expansion.

These results suggest three ice sheet volume regimes: a small ice sheet (<25 m.s.l.e.) which easily melts; a medium
ice sheet that can grow rapidly through temperature-elevation feedbacks and the merging of the Cordilleran and Laurentide
(>25, <60 m.s.l.e.); and a large ice sheet (>60 m.s.l.e.) which is sensitive to a change in climate due to strong positive melt
feedbacks, such as the melt-elevation feedback, melt-albedo feedback and the formation of proglacial lakes (see Scherrenberg
et al., 2024) and a thermodynamical decoupling (Bintanja and van de Wal, 2008). A sharp increase in $CO_2$, combined with the
strong summer insolation at glacial terminations could then further accelerate the melt of the ice sheets. A successfully
modelled termination therefore hinges on whether this melt-feedback loop is triggered. If the ice sheet falls within the large
regime, a period with strong summer insolation can more easily trigger the melt feedback loop. $CO_2$ and insolation conditions
for which a small or medium-sized ice sheet could survive, can yield a full collapse of a large ice sheet. Reversely, if the
combination of $CO_2$, insolation and ice volume fails to trigger strong enough melt-feedback processes, the ice sheets do not
fully melt, prolonging the glacial period.

### 3.3 Disentangling $CO_2$ and insolation

Our baseline simulation can capture the frequency change during the MPT using prescribed changes in $CO_2$ and summer
insolation. In this section, we explore the model response if either one is removed.

We conduct four constant insolation simulations by applying the constant insolation at 0 kyr ago, (implying a constant
caloric summer insolation of ~5.8 GJ/m$^2$), 5 kyr ago ("enhanced" insolation; ~5.9 GJ/m$^2$), 25 kyr ago (an insolation minimum;
~5.4 GJ/m$^2$), and 10 kyr ago (an insolation maximum; ~6.1 GJ/m$^2$). Note that the caloric summer insolation at 0 kyr ago is
close to the mean of the past 1.5 million years (5.84 GJ/yr). Each constant insolation simulation has the same set-up as the
baseline, but we apply a constant (monthly varying) insolation. Temperature change results only from $CO_2$ and the albedo

feedback. Note that the resulting glacial periodicity can still match orbital cycles, as past $CO_2$ levels were not independent from insolation.

Time-series of modelled sea level and benthic $\delta^{18}O$ in the constant insolation and baseline simulations are shown in
Fig. 7. Three out of four constant insolation experiments do not capture the Early Pleistocene glacial cycles, while the constant_insolation_5kyr_ago matches the glacial-interglacial periodicity during the Pleistocene, though with long interglacial periods during the Late Pleistocene.

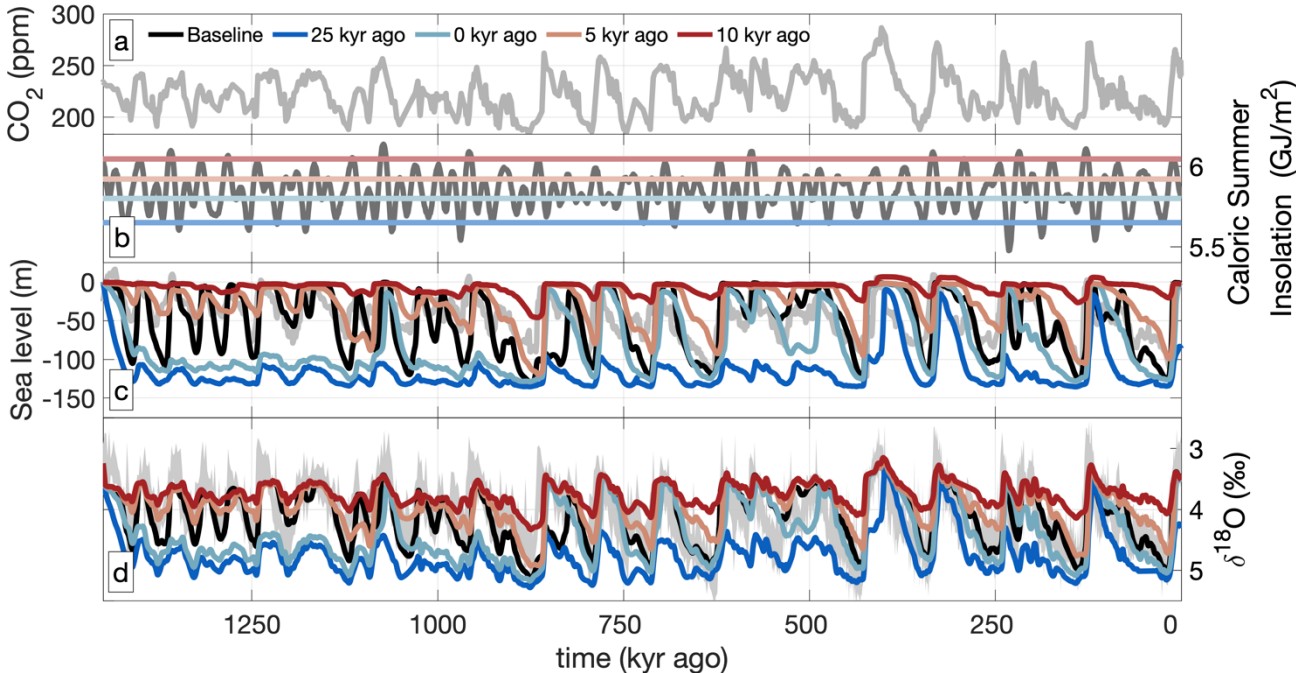

**Figure 7.** Time-series of prescribed $CO_2$ (a), insolation (b), sea level (c) and $\delta^{18}O$ (d). $CO_2$ reconstruction by Yamamoto et al. (2022; a), sea-
level by Rohling et al. (2012; c), and $\delta^{18}O$ Ahn et al (2017; d) all shown in grey.

Whether the constant insolation simulations match the periodicity depends on interglacial $CO_2$ levels and the constant insolation strength. In the Yamamoto et al. (2022) record, interglacial $CO_2$ levels are low during the Early Pleistocene (~240-250 ppm) and increase during the Late Pleistocene, with high interglacial $CO_2$ levels during the past 400 kyr (~260-270 ppm). If summer insolation is weak (constant_insolation_0kyr_ago/25kyr_ago), only some Late Pleistocene cycles are captured. If
summer insolation is relatively strong (constant_insolation_5kyr_ago), the low interglacial $CO_2$ levels during the Early Pleistocene can trigger terminations. Very strong summer insolation (constant_insolation_10kyr_ago) prevents the growth of ice and the ice volume stays below the "small" threshold regime.

To investigate if orbital cycles alone can capture the MPT in our set-up, we conduct simulations with constant $CO_2$ instead. As such, there is no change in $CO_2$ concentration for glacial, interstadial or interglacial periods. Fig. 8b shows sea
level time-series of simulations forced by constant 240, 220 and 210 ppm $CO_2$ concentrations (constant $CO_2$ experiments).

The modelled ice volume differs substantially between these three simulations: Constant_CO2_240 yields small ice volumes (generally less than 50 m.s.l.e.), which is around half of that in the constant_CO2_220 and constant_CO2_210 simulations. The ice sheets in the constant_CO2_240 simulation mostly melts at 41-kyr intervals. The constant_CO2_220 largely follows the glacial-interglacial periodicity during the Early Pleistocene and also captures some of the termination events and prolonged glacial periods during the Late Pleistocene. Constant_CO2_210 has an even lower $CO_2$ concentration, leading to long glacial cycles. In all these simulations, the ice sheets can still fully melt, despite low constant $CO_2$ concentrations. This is facilitated by positive melt feedbacks combined with strong summer insolation.

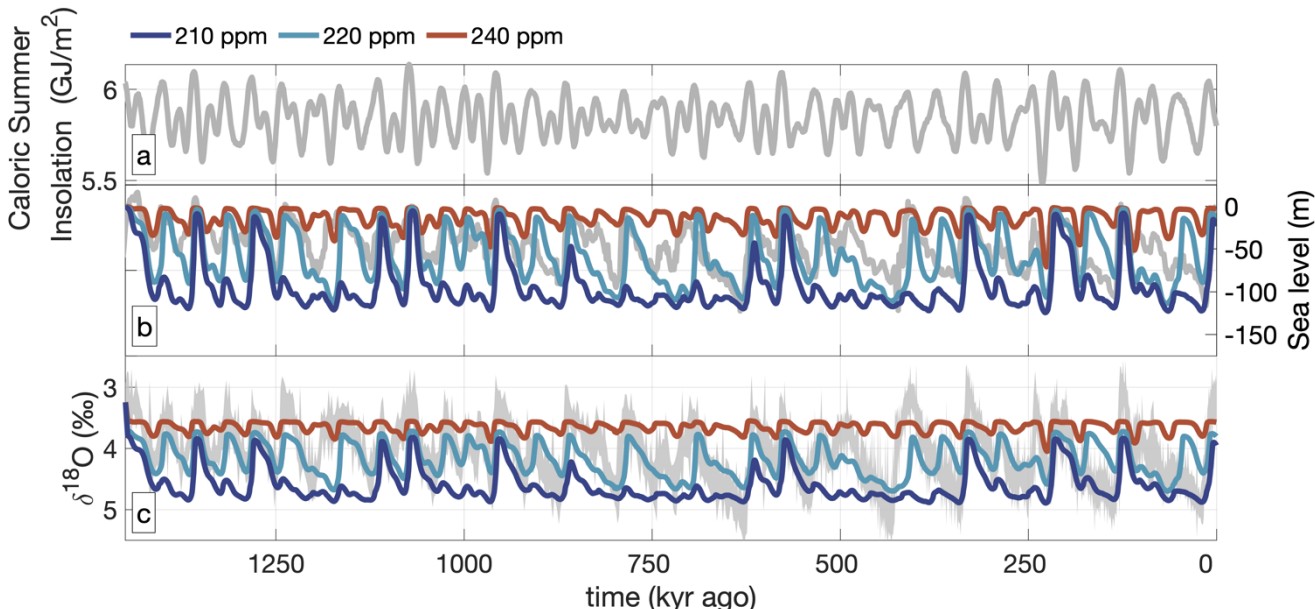

**Figure 8:** Time-series of caloric summer half-year insolation (a), sea level (b), and $\delta^{18}O$ (c) of the experiments with constant 240 ppm (red), 220 ppm (cyan) and 210 ppm (blue) $CO_2$ levels. Observed sea level (Spratt and Lisiecki, 2016; b) and $\delta^{18}O$ (Ahn et al., 2017; c) is shown in grey. Note that the $\delta^{18}O$ of sea water is calculated using $CO_2$, and all variability of $\delta^{18}O$ in the constant_CO2 simulations therefore results from the ice sheets.

Fig. 9 shows wavelet transforms of the sea level and compares it to the summer insolation, which represents the only driver of temperature change in the constant $CO_2$ simulations. Constant_CO2_240 and constant_CO2_220 simulations show 41-kyr periodicity during the Early Pleistocene, but this generally persists into the Late-Pleistocene. While constant_CO2_220 and constant_CO2_210 show increased periodicity when insolation maxima are skipped, neither maintain a prolonged 100-kyr periodicity.

While precession is present in the climate forcing (Fig. 9d), the ~20-kyr signal is mostly absent in the frequency spectrum of the modelled sea level. Terminations are initiated when insolation is strong enough to initiate positive melt feedbacks. The insolation threshold for a termination depends also on $CO_2$, as the constant_CO2_240, 220 and 210 simulations generally melt at a caloric summer half-year insolation of roughly 5.9 $GJ/m^2$, 6.0 $GJ/m^2$ and 6.1 $GJ/m^2$ respectively. The

terminations tend to occur during peaks in the caloric insolation (generally obliquity), while many smaller peaks (mostly precession) are skipped. Additionally, we have used caloric summer insolation as a driver for temperature change, which

accounts for a change in the duration of the melt season that is caused by, and partly compensates the precession signal (see Huybers, 2006). Terminations will therefore tend to correlate to obliquity maxima and filter out precession. The resulting ice volume frequency spectrum will then be dominated by the successful terminations.

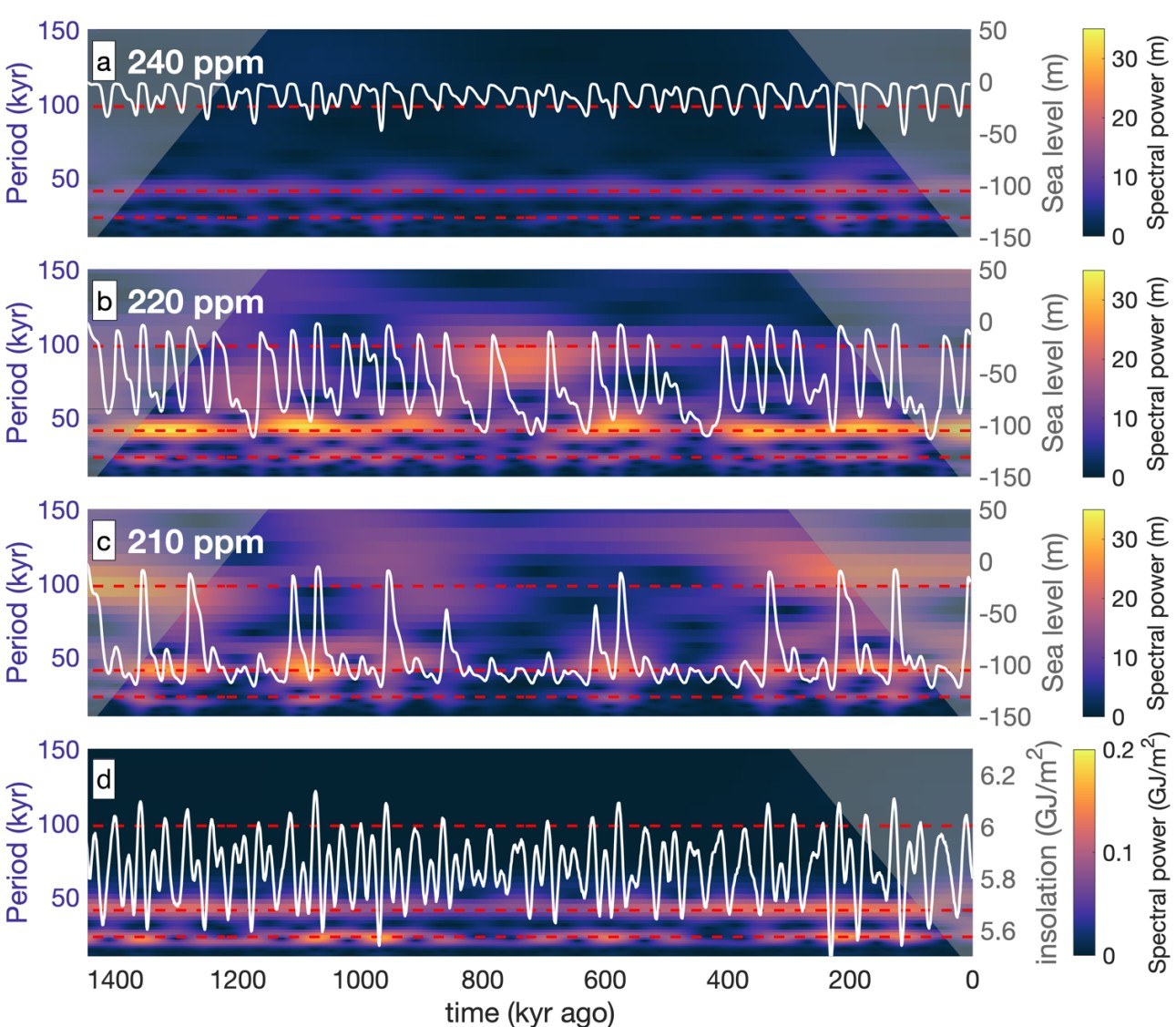

**Figure 9.** Wavelets showing the frequency in the sea level of 240 ppm (a), 220 ppm (b) and 210 ppm (c), compared to caloric summer half-
345 year insolation (d). Corresponding time-series are shown in white.

## 4. Discussion

In this study we simulate the 1.5-million-year Northern Hemisphere ice-sheet evolution using climate forcing driven by prescribed $CO_2$ and insolation, and implicit ice-sheet-climate interactions. Using these driving forces, we are able to capture the glacial-cycle frequency change at the MPT without any change in model-set-up or basal friction. Our modelled MPT results mainly from the prescribed $CO_2$ record: In the Late Pleistocene, low interstadial $CO_2$ levels are maintained throughout some insolation maxima, prolonging the glacial period.

While we do simulate the frequency change and Late Pleistocene ice volume, we only obtain a small shift in the sea level amplitude across the MPT. Our simulations are forced by and dependent on the leaf-wax proxy $CO_2$ record by Yamamoto et al. (2022), which is currently the only continuous $CO_2$ record of the past 1.5 million years. This record has relatively low Early Pleistocene (inter)glacial $CO_2$ levels compared to boron-isotope-based records (e.g., see Chalk et al. 2017) and carbon cycle modelling results (e.g., Willeit et al., 2019). If Early Pleistocene $CO_2$ concentrations are indeed underestimated in the Yamamoto et al. (2022) record, this would lead to temperatures that are too low, and thereby generate a too large ice volume amplitude. Additionally, we applied a constant (present-day) sediment map, leading to too-high friction and consequentially too-large ice volume during the Early Pleistocene. However, we also show that ice volume influences the termination events, and a different ice sheet size may collapse at different $CO_2$ and insolation intensity. As such, our lack of substantial amplitude change may influence our results.

Our North American ice volume shows a threshold regime: Small ice sheets melt easily. Medium ice sheets are less sensitive due to their location combined with the merging of the Cordilleran and Laurentide ice sheet inducing a positive feedback, in agreement with Bintanja and van de Wal (2008). Large ice sheets have a long southern margin and are sensitive to small increases in temperature. A successful termination hinges on whether the climate forcing ($CO_2$ and insolation) and ice volume can trigger a strong melt feedback, which is facilitated by melt-elevation feedback and proglacial lakes. These threshold regimes are in line with several studies that suggest that the Late Pleistocene terminations only take place if ice volume is large enough (Parrenin and Paillard, 2003; Bintanja and van de Wal 2008; Abe-Ouchi et al., 2013; Verbitsky et al., 2018; Berends et al., 2021a). Additionally, several studies (see Parranin and Paillard, 2003; Berends et al., 2021a; Legrain et al., 2023) used conceptual models to show that such a threshold behaviour could lead to a change in glacial-interglacial periodicity, here we show the same for a more realistic bed topography and climate.

This threshold behaviour may also explain why the Late Pleistocene glacial terminations only take place during some, but not all, insolation maxima. Low interstadial $CO_2$ levels may cause an insolation maximum to be skipped if the ice-sheet is medium-sized, but even lower glacial $CO_2$ concentrations could potentially still generate a full collapse if the ice sheet is large-sized. The periodicity of the benthic $\delta^{18}O$ record is then dominated by the successful terminations, which only occur when the combination of insolation, $CO_2$ and ice volume are able to trigger a strong enough melt-feedback loop.

These ideas are further explored by simulations that use constant $CO_2$ levels or insolation. Using constant insolation, we are able to generate the terminations of the Early and Late Pleistocene, though whether these glacial cycles are captured is

conditional to a narrow range of constant caloric summer insolation combined with the prescribed interglacial $CO_2$ levels. If

interglacial $CO_2$ is high (Late Pleistocene), terminations are possible at lower constant summer insolation. If interglacial $CO_2$ is low (Early Pleistocene, according to Yamamoto et al., 2022), stronger constant summer insolation is needed for terminations.

If we used constant $CO_2$ instead, we could capture the 41 kyr cycles of the Early Pleistocene. Using low constant $CO_2$ levels (220 ppm), we could capture the Early Pleistocene cycles and some Late Pleistocene glacial cycles. However, no persistent 100 kyr periodicity was established. This is partially in line with conceptual model results by Legrain et al., (2023),

finding that orbital cycles alone can capture some (but not all) characteristics of the MPT. However, if the carbon cycle was modelled and $CO_2$ was allowed to freely evolve, the modelled glacial-interglacial frequency could be different. For example, an intermediate complexity model simulation with active carbon cycle component by Willeit et al. (2019) generated a persistent 100-kyr periodicity with only orbital forcing.

The climate forcing of the constant_CO2 simulations is driven by insolation, which encloses a precession signal. This

signal is filtered from the modelled ice volume as deglaciations are only triggered if summer insolation is strong, which tends to filter out the relatively weaker insolation maxima (induced by precession). The resulting ice volume frequency is therefore dominated by obliquity, as only these trigger successful terminations in the Early Pleistocene. This idea, which proposes that a threshold in caloric summer insolation can generate precession cancellation, has also been suggested by Tzedakis et al. (2017). In this study, we have found that the non-linear response of the Northern Hemisphere ice sheet towards climate forcing

can filter out the precession signal.

We also found a very strong sensitivity to the constant $CO_2$ levels, as a decrease from 240 to 220 ppm $CO_2$ yields a doubling in ice volume. While it is uncertain if this sensitivity fully holds-up in a fully-coupled Earth-System Model set-up, it does partially agree with similar experiments using an intermediate complexity model (see Ganopolski and Calov, 2011). However, their 240 ppm $CO_2$ level already shows some skipped terminations at obliquity maxima, while our simulation yields

a persistent 41-kyr periodicity in our simulation instead.

This high sensitivity to $CO_2$ also highlights the importance of accurate $CO_2$ reconstructions to detect the changes in long-term $CO_2$ concentration over the MPT. We have used a leaf-wax record (Yamamoto et al., 2022) that has (indirectly) reconstructed $CO_2$, which was calibrated to (directly measured) $CO_2$ record obtained from air trapped inside ice (Bereiter et al., 2015). However, the leaf-wax record has larger uncertainties compared to ice cores, and as such, $CO_2$ records before the

405 ice core (800 kyr ago) is still uncertain. Nevertheless, the shift from 41 to 100 kyr periodicity in the $CO_2$ record is also found in boron-based $CO_2$ reconstructions (e.g., Dyez et al., 2018). Therefore, the frequency shift in $CO_2$ is consistent among different $CO_2$ records. Our Early Pleistocene from our baseline simulation results generally agree with Watanabe et al. (2023), who found based on ice-sheet model simulations that the Early Pleistocene glacial cycles follow from orbital oscillations, with minimal effect from $CO_2$. We were however able to capture the Early Pleistocene cycles using either constant $CO_2$ levels or a

410 narrow range of constant insolation.

Simulating 1.5 million years requires a trade-off between explicitly modelling processes and computational time. While we simulate some ice-sheet climate interactions, they are more complex in reality. For example, we do not include any

feedback from the ocean circulation or sea ice. Another limitation of our approach is that the climate forcing is only based on just two ensemble-mean climate time-slices. This choice was made due to the large differences between climate models, whereas the PMIP4 ensemble shows good results (see Kageyama et al., 2021). These GCM simulations were conducted with a prescribed LGM ice sheet, and the high albedo of the ice sheet leaves a cold imprint on the temperature field and creates a large temperature gradient between ice and ice-free areas. If the ice volume is close to LGM, the extent will therefore be close as well. We do not see this as a limitation, as our main goal here is not to make accurate reconstructions, but rather focus on the long-term ice volume change and the processes behind the change in glacial cycle periodicity.

The simulated MPT, and much of the glacial-interglacial variability of the past 1.5 million years results from a change in the prescribed $CO_2$ forcing, but the origin of the amplitude and frequency in the $CO_2$ record remains uncertain and was not studied here. As we use prescribed $CO_2$, (failed) terminations are already present in the $CO_2$ record: High $CO_2$ levels could be a symptom of successful terminations, while low $CO_2$ levels could be a symptom of unsuccessful terminations. Our model then replicates these failed and successful terminations. However, even with constant $CO_2$ levels, we can simulate many terminations during the Late Pleistocene, even though the overall periodicity does not match. A low constant $CO_2$ concentration (210 ppm) does skip several of these terminations. This suggest that the melt of the ice sheet could have been prompted by insolation, but it has been enhanced by an increase in $CO_2$. However, our simulations do not explain the change in frequency and amplitude present in the $CO_2$ record, which involves ice-sheet-carbon-cycle interactions. To conclude, our results suggest that $CO_2$ can have a key role in the MPT, and that maintaining low $CO_2$ levels during insolation maxima could lengthen glacial cycles. However, to truly uncover the origin of the MPT will require a coupled ice-sheet-climate-carbon-cycle model.

**5. Conclusions**

In this study, we simulate the Northern Hemisphere ice sheet evolution during the past 1.5 million years using an ice-sheet model forced by prescribed $CO_2$ and insolation forcing and implicit climate-ice-sheet interactions. Our main goal is to simulate and investigate the frequency change during the MPT. Additionally, we investigate the separate contribution from $CO_2$ and insolation by conducting experiments using constant insolation or constant $CO_2$. Our main conclusions are:

- Our experiments suggest that the MPT can be simulated by prescribing only $CO_2$ and summer insolation as forcing, though both $CO_2$ and insolation are necessarily to fully capture the frequencies of the system over the past 1.5 million years.
- The size of the ice sheet itself affects its stability, and the $CO_2$ levels and insolation thresholds at which it melts. We show three "threshold regimes" for the North American ice sheet: A small North American ice sheet that melts at relatively warm climates. A medium sized ice sheet that is less sensitive towards collapse as it can grow towards (high latitude) central Canada, where the Cordilleran and Laurentide ice sheets merge, converging ice flows and merging ablation zones. The "large" ice sheets can only thicken or grow towards the warmer South, reducing growth rates. These large ice sheets can easily melt through positive melt feedbacks, such as the ice albedo, melt-elevation feedbacks and the creation of proglacial lakes.

- Deglaciations at insolation maxima may be skipped if $CO_2$ levels are low. If $CO_2$ remains low during an insolation maximum, the threshold at which feedback processes (e.g., albedo and melt-elevation interactions, and proglacial lakes) trigger a self-sustained collapse of the ice sheet is not reached, prolonging the glacial cycle. Thus, maintaining low $CO_2$ during insolation maxima may increase glacial cycle periodicity.

- In agreement with reconstructions, none of our simulations capture a strong precession signal, even though precession is present in the insolation forcing. Caloric summer insolation maxima due to precession are weaker compared to obliquity. As such, the ice sheets are less likely to melt at precession maxima. The modelled ice volume frequency spectrum is then dominated by successful terminations, coinciding with strong insolation maxima (obliquity) and thus filtering out precession.

In this study, $CO_2$ is used as model forcing, while in reality $CO_2$ is also a feedback resulting from the complex interactions in the carbon cycle. Our results show that $CO_2$ can play a key role in the MPT, but to unravel the mechanisms of the MPT in more detail requires a model not only with ice, climate and prescribed $CO_2$, but also with an explicit carbon cycle.

## Appendix A: Climate forcing

The ice sheet is forced with transient changing precipitation and temperature forcing. To reduce computational resources compared to coupled ice-climate set-ups, we interpolate between pre-calculated LGM and PI climates using a matrix method (see Berends et al., 2018; Scherrenberg et al., 2023). For the monthly ($mnth$) temperature forcing ($T$) at each grid-cell ($x, y$), we use the following linear interpolation:

$$T(x, y, mnth) = w_T(x, y)\, T_{PI}(x, y, mnth) + \left(1 - w_T(x, y)\right) T_{LGM}(x, y, mnth). \tag{A1}$$

$w_T$ represents the interpolation weight and depends on external forcing ($w_e$) and an albedo feedback ($w_a$). We allow some extrapolation for colder than LGM or warmer than present-day climates, though each interpolation weight is capped between -0.5 and 1.5.

The external forcing interpolation weight depends on $CO_2$ ($CO2$; obtained from Yamamoto et al., 2022 or Bereiter et al., 2015) and caloric summer half-year insolation at 65°N ($Q_{65°N}$; Tzedakis et al., 2017). We use the following equation to

470 calculate the weight from the prescribed forcing:

$$w_e = \frac{CO2 - 190\ \text{ppm}}{280\ \text{ppm} - 190\ \text{ppm}} + \frac{Q_{65°N} - 5.8\ GJ/m^2}{0.55\ GJ/m^2}. \tag{A2}$$

This ratio was obtained by conducting a preliminary experiment based on de Boer et al. (2013) and Berends et al. (2021b), where we modify $w_e$ to obtain good agreement with benthic $\delta^{18}O$ from Ahn et al., (2017). We then fitted the resulting $w_e$ to $CO_2$ and insolation and fine-tuned it to obtain Eq. (A2).

To calculate $w_a$ (the albedo feedback), we first calculate 2D fields of the amount of insolation that is absorbed by the surface ($I$). This depends on surface albedo ($\alpha_s$) and the grid-cells insolation at the top of the atmosphere ($Q$):

$$I(x,y) = \sum_{m=1}^{12} Q(x,y,mnth)\left(1 - \alpha_s(x,y,mnth)\right). \tag{A3}$$

The albedo is generated by the ice-sheet model. First a background albedo is applied based on the ice (0.5), land (0.2) and ocean (0.1) surfaces. We then add a layer of snow that can increase albedo to up to 0.85. Using masks, climate and insolation from PI and LGM, we calculate absorbed insolation fields for the climate time-slices as well ($I_{PI}$ and $I_{LGM}$). Using these three fields, we calculate a local interpolation weight for absorbed insolation ($w_i$):

$$w_i(x,y) = (I(x,y) - I_{LGM}(x,y)) \,/\, (I_{PI}(x,y) - I_{LGM}(x,y))\,. \tag{A4}$$

Albedo has both a local and regional effect. We apply a gaussian smoothing of 200 km on $w_i$ to obtain $w_{i,smooth}$. We also calculate a domain-average $w_i$, which is $w_{i,domain}$. These three interpolation fields are then combined to obtain the albedo feedback, following the approach by Berends et al. (2018):

$$w_a(x,y) = \frac{w_i(x,y) + 3\,w_{i,smooth}(x,y) \; + \; 3\,w_{i,domain}(x,y)}{7}. \tag{A5}$$

The albedo interpolation weight is then combined with the external forcing to obtain the $w_T$ from Eq. (A1):

$$w_T(x,y) = \frac{3\,w_e(x,y) + w_a(x,y)}{4}. \tag{A6}$$

As such, temperature depends on $CO_2$, caloric summer half-year insolation and a spatially varying albedo/insolation field.

To interpolate precipitation ($P$), we use the following equation:

$$P = exp\begin{pmatrix}(1 - w_P(x,y)\,)\log\left(P_{PI}(x,y,mnth)\right)\\ + w_P(x,y)\log\left(P_{LGM}(x,y,mnth)\right)\end{pmatrix}. \tag{A7}$$

$w_P$ is the interpolation weight for precipitation, and is calculated with respect to local and domain-wide topography changes, reflecting changes in atmospheric circulation and the dry climates on top of ice domes. First, we calculate the total change in topography ($s$) in the domain with respect to PI and LGM:

$$w_{s,domain} = \frac{\sum s - \sum s_{PI}}{\sum s_{LGM} - \sum s_{PI}} \tag{A8}$$

$w_{s,domain}$ is the interpolation weight from a domain-wide change in topography. If a grid-cell has ice during the LGM, and thus a large change in topography, we also interpolate with the local topography change:

$$w_{s,local}(x,y) = \frac{S(x,y) - S_{PI}(x,y)}{S_{LGM}(x,y) - S_{PI}(x,y)}\,w_{s,domain}(x,y). \tag{A9}$$

However, if a grid cell had ice during neither PI nor LGM, $w_{s,local}$ is equal to $w_{s,domain}$. To obtain the final interpolation weight ($w_P$), we combine the local and domain-wide topography change:

$$w_P(x,y) = w_{s,local}(x,y)\,w_{s,domain}(x,y). \tag{A10}$$

## Appendix B: δ¹⁸O model

IMAU-ICE includes a benthic δ¹⁸O routine, which calculates the δ¹⁸O contribution from ice volume and deep-water temperature. This method is based on de Boer et al. (2013) and Berends et al. (2021b). Deep water temperature change (ΔTd) is based on $CO_2$ levels ($CO2$):

$$\Delta \text{Td} = \left(\frac{280 \text{ ppm} - CO2}{280 \text{ ppm} - 190 \text{ ppm}}\right) 2.5°C \qquad . \qquad (\text{B1})$$

A 3000 kyr running-mean is applied to reflect the lag between atmospheric and deep-ocean temperatures. We then multiply this by 0.28 to obtain the δ¹⁸O contribution from deep-water temperature.

For the ice sheets, we calculate a δ¹⁸O contribution for every grid-cell ($I$). We calculate a δ¹⁸O of snow accumulation based on Clarke et al. (2005):

$$I(x,y) = I_{ref}(x,y) + 0.35(T(x,y) - T_{PI}(x,y) - \gamma(s(x,y) - s_{PI}(x,y)) - 0.0062(s(x,y) - s_{PI}(x,y)). \qquad (\text{B2})$$

Here, T represents the annual mean temperature, and $s$ represents the surface topography. The total contribution from each ice-sheet is added together and multiplied by 1.1 to reflect that we do not simulate Antarctica. $I_{ref}$, which is the (present-day) reference isotope concentrations, is calculated using the following parameterization by Zwally and Giovinetto (1997):

$$I_{ref}(x,y) = 0.691 * T(x,y) - 202.172. \qquad (\text{B3})$$

We then add the deep-water and ice-sheet contributions together to obtain the benthic δ¹⁸O.

*Code availability:* The ice-sheet model IMAU-ICE is described by Berends et al. (2022). The model version used in this study, as well as configuration files are available on Zenodo [DOI will be added upon acceptation]. To conduct the simulations, additional files are required. These include the prescribed $CO_2$ (see Bereiter et al., 2015 and Yamamoto et al., 2022), climate forcing (PMIP4 database: https://esgf-node.ipsl.upmc.fr/search/cmip6-ipsl/, last access: 16 Aug 2024), insolation (Laskar et al., 2004, Tzedakis et al., 2017), initial topography (ETOPO: https://doi.org/10.7289/V5C8276M, Amante and Eakins, 2009; BedMachine: https://doi.org/10.5067/5XKQD5Y5V3VN, NSIDC, 2024), and basal friction (Gowan et al., 2019 and Laske and Masters, 1997). For more information, contact the corresponding author.

*Data availability:* The results are available in a 2 kyr (2D fields) and 100-year (scalar) output frequency at Zenodo [DOI will be added upon acceptation]. Additional 2D fields can be requested by contacting the corresponding author.

*Author contributions.* MS conducted the simulations and has written the manuscript. The set-up for the experiments was created by RW, CB and MS. CB provided model support. All authors have provided input to the manuscript and analysis of the results.

*Competing interest.* The authors declare that they have no conflict of interest.

*Acknowledgements.* The Dutch Research Council (NWO) Exact and Natural Sciences supported the supercomputer facilities for the Dutch National Supercomputer Snellius. We would like to acknowledge the support of SurfSara Computing and Networking Services.

*Financial support.* M.D.W. Scherrenberg is supported by the Netherlands Earth System Science Centre (NESSC), which is financially supported by the Ministry of Education, Culture and Science (OCW) on grant no. 024.002.001. C.J. Berends is funded by the NWO under grant no. OCENW.KLEIN.515.

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
