# Peer review of "CO2 and summer insolation as drivers for the Mid-Pleistocene transition"

_Climate of the Past, 2024_

## Author Response (AR1)

First of all, we would like to thank the reviewer for their review of our manuscript, which has helped us to address issues and improve our conclusions. In this rebuttal, we would like to address their concerns. The reviewers' comments are shown in bold, while our answers are shown in regular font-type.

**Review of Scherrenberg et al., "CO2 and summer insolation as drivers for the Mid-Pleistocene transition," submitted to Climate of the Past.**

The authors present modeling results of the Mid-Pleistocene Transition using a simple iceclimate model. This is a key approach for exploring the mechanisms behind this major climate transition, which remains an unsolved question in paleoclimatology. Their model is forced with caloric summer insolation, and an indirect atmospheric CO2 reconstruction derived from leaf wax spanning the last 1.5 Ma. In addition to baseline simulations, the authors conducted additional simulations with either constant atmospheric CO2 concentrations or constant insolation to disentangle the impact of each forcing on the climatic cycles of Pleistocene.

Overall, the manuscript is well-written and structured, although the second part is somewhat challenging to follow due to a mix of the results and discussion. The figures are well chosen and informative but would benefit from more specific description rather than broader summaries. I do have some questions about the way the authors interpret their results. Specifically, I find the boundary between the conclusions derived from their baseline simulations, the ones derived from their constant\_ simulations, and the assumption based on paleoclimatic records very blurred. In addition, the use of the Yamamoto's CO2 record as a model forcing, that did not register any glacial CO2 decrease through MPT, is not obvious to link with one of the main conclusions of the authors (i.e. the role of glacial CO2 concentrations in the trigger of the MPT).

**Major comments:**

(1) The authors used the Yamamoto CO2 record as forcing for their model. As it is the only continuous CO2 record through the MPT, this choice is perfectly understandable. Nevertheless, the results derived from the model are per definition dependent on this record and this should be mentioned. A second point, probably the most important one, is the interpretation made of this record by the authors: "Yamamoto et al. (2022) (...) find a decrease in glacial CO2 concentrations across the MPT". I strongly disagree with this statement. Surprisingly, Yamamoto et al. found constant CO2 glacial concentrations through the MPT, and a gradual increase in interglacial CO2 concentrations. All along the paper, the authors suggests that a glacial decrease in CO2 concentrations would have skipped some deglaciations during the MPT and the late Pleistocene: the baseline experiment performed using the Yamamoto's record seems to demonstrate that it is not necessary to involve a decrease in glacial CO2 concentrations to « reproduce » the MPT. It would be interesting if the authors could clarify the relation between the use of Yamamoto's record as a forcing and their conclusion that a decrease in glacial CO2 concentrations would have

To address the first point: We indeed chose the Yamamoto et al. (2022) record, as this is currently the only continuous  $CO_2$  record that covers the past 1.5-million. We have now addressed this in the introduction (see line 89).

The Yamamoto record does have significantly lower Early Pleistocene glacial/interglacial  $CO_2$  levels compared to both proxies and carbon cycle modelling (e.g., Chalk et al. 2017; Willeit et al., 2019). We have now addressed this in the discussion section at line 344-345. These low  $CO_2$  concentrations may also partially explain our too large Early Pleistocene ice volumes. In the discussion section, we now explicitly address that our results are indeed dependent on the Yamamoto et al. 2022  $CO_2$  record (See line: 344-349).

To address the second point, we made several changes to the manuscript to improve the support to the main conclusions:

In our simulations we found that if  $CO_2$  and insolation are strong, it can trigger a termination. This threshold also depends on the ice sheet, as a larger ice sheet is more prone to collapse when temperatures rise. Insolation maxima fail to trigger terminations if low  $CO_2$  levels are maintained throughout the interglacial period.

In the Early Pleistocene, obliquity maxima tend to coincide with relatively high CO2 levels (~240-250 ppm), allowing for terminations. However, in the Late Pleistocene, low interstadial CO2 levels are sometimes maintained through insolation maxima, preventing deglaciation. This phenomenon is why we referred to "*decrease in glacial CO*2", but we followed the reviewers' suggestions to rephrase this line to "*lowering interstadial CO*2" (e.g., see line 363; line 425), or more specifically explain that deglaciation is prevented if low CO2 levels are maintained throughout the insolation maxima (see lines 341-344). Additionally, the constant\_CO2 experiments also show that maintaining a low CO2 level throughout insolation maxima can increase glacial cycle periodicity.

We removed the erroneous statement "Yamamoto et al. (2022) (...) find a decrease in glacial CO2 concentrations across the MPT", revised the discussion and results sections to more clearly explain the aforementioned model behaviour.

(2) This second point is related to the first one: it is unclear which conclusion is derived from their baseline simulation, from their constant\_ simulations or from literature. For instance, in the case of the  $CO_2$ , the two approaches are opposite: in the baseline simulation, the authors forced their model with a glacial-constant  $CO_2$  record while the comparison of constant\_simulation would indeed allow to discuss the decrease of glacial  $CO_2$  concentrations (but not only), even if these experiments are theoretical.

The manuscript benefits from a clearer distinction between the proxy-forced and theoretically-forced simulations. We revised and restructure the discussion section to improve these conclusions.

First of all, we made a clearer distinction whether a statement or conclusion is derived from the proxy  $CO_2$  record, the constant\_ $CO_2$ /insolation simulations or literature. For example, we now state "low constant  $CO_2$  levels" rather than "low  $CO_2$  levels" when discussing the

constant\_CO2 experiments; see e.g., line 372. Similarly, we specify each (relevant) mention of CO2 whether it concerns interglacial/glacial or interstadial levels.

Secondly, we rewrote and restructured parts of the discussion section (see 339-400). Parts of the results section were removed or moved to the discussion section and rewritten, as these contributed to the unclear support to our conclusions. We now explicitly state that the constant\_CO2 experiments do not make a distinction between glacial and interglacial  $CO_2$  levels (see line 306). We made a clearer distinction between the conclusions derived from the baseline simulation, the constant\_CO2 and constant\_insolation experiments in both abstract (er.g., see 25-28) and discussion section (see 367-385).

(3) The results derived from the constant\_insolation are interesting. Nevertheless, I wonder to what extent these results (the absence of climatic cycles in the pre-MPT world) depend on the chosen insolation value (set as present-day by the authors). A comparison of different insolation values, similar to what was done for the CO2\_constant experiments, would strengthen these conclusions.

For the revised manuscript, we added three additional simulations. Firstly, the constant\_insolation\_10kyr\_ago (a Northern Hemisphere insolation maximum), which has glacial cycles with small ice volume amplitude. Secondly, we added the constant\_insolation\_25kyr\_ago (an insolation minimum) which has very long glacial periods.

Lastly, we added the constant\_insolation\_5kyr\_ago. This simulation has a slight increase in constant caloric summer insolation compared to the present-day insolation, but captures all major termination events during the past 1.5 million years.

The constant\_insolation\_5kyr\_ago generates the periodicity and amplitude of the Early Pleistocene. It does reasonably well for the Late Pleistocene as well, though has relatively long interglacial periods and a slightly reduced glacial-interglacial amplitude compared to sea level reconstructions.

We therefore remove the conclusion "Early Pleistocene are dominated by orbital cycles, while late Pleistocene is dominated by  $CO_2$ ", as it is possible to mostly capture the full glacial-interglacial periodicity of the past 1.5 million years with just prescribed  $CO_2$ . Though, note that past  $CO_2$  levels were (indirectly) affected by long-term insolation changes, thus the ice volume evolution of  $CO_2$ -only-forced simulations can still match orbital cycle periodicity. This is now also explicitly stated in line 293.

We added these three simulations to figure 6 and figure 7. We also changed the description of the constant\_insolation simulations to accommodate these new simulations (see line 288-304).

(4) The authors have chosen to use a constant sediment map throughout the MPT, thereby excluding the possibility of testing the regolith hypothesis. While I understand this decision, as testing all MPT hypotheses is challenging and time-consuming, I strongly suggest that the authors exercise greater caution in interpreting their results : "Using these driving forces, we are able to capture the MPT without any change in (...) basal friction". Simulating the MPT is not a binary process where one either succeeds or fails. While I

agree that their simulations effectively capture the change in frequency, there is no guarantee that adding a variable sediment mask wouldn't improve the simulations. In my view, if the study still does not include a varying sediment mask, it cannot address the regolith hypothesis.

We have recently conducted the "sediment\_change" simulation, which has reduced "sediment" friction during the Early Pleistocene (1.5 - 0.8 million years ago). In the sediment\_change simulation, we replaced the basal friction map with a new map that treats the entire domain as if it is covered by sediment. This simulation has a ~10% reduction in amplitude during the Early Pleistocene, mostly due to a lower average thickness of the ice sheets. Additionally, the termination at MIS 21 (~865 kyr ago), which is skipped in the baseline, is modelled when using lower friction. This is because the lower friction makes the ice sheet more likely to collapse at insolation maxima (which is of course the method behind the regolith hypothesis). This sediment\_change simulation was added to the supplementary information, and introduced in lines 214-216.

We do believe we should mention that we were able to model the frequency change with only CO2 and insolation, though also highlight that the model results improve slightly when applying lower basal friction. Though is now addressed in the discussion section (see lines 350-351).

(5) I found the organization of the paper, particularly paragraph 3.2 in the results section, somewhat confusing. The discussion within the results section also contributed to this confusion. For example, the section concludes with one of the study's most significant findings: "A gradual decrease in glacial CO2 levels could therefore explain the MPT." This conclusion is not a direct observation of the result. As a result, I believe the discussion section is somewhat brief and would have benefited from a more thorough comparison of the authors' results with recent studies that have also aimed to simulate the MPT. For instance, Willeit et al. (2019) discuss the role of CO2 using multiple scenarios of unconstrained CO2 forcing; Legrain et al. (2023) draw similar conclusions to the authors using conceptual modeling; and Verbitsky et al. (2018) present interesting findings based on a physical approach of the MPT.

We rewrote, restructured and expanded the discussion section. Several lines of paragraph 3.2 were moved to / merged with the discussion section. To give a brief summary of the changes to the results section:

- Lines 221-224 (of the submitted version) were be removed. In these lines, we address our lack of an active carbon cycle, which was already addressed in the discussion section.
- At the end of section 3.2, we explained how CO2, insolation and ice sheets relate to glacial-interglacial periodicity. This was revised and moved to the discussion (see lines 339-343, and 352-366).
- The two sentences at line 282-284 (submitted version) were removed.
- Lines 303-304 (submitted version) were removed.
- In section 3.2, we compared our precession cancelling findings to Tzedakis et al. (2017). These statements were revised and moved to the discussion section (see lines 379-385).

Lastly, we added comparisons with Legrain et al (2023), Verbitsky et al (2018) and Willeit et al (2019) to the revised manuscript (see lines 374-378; 356-361). The former two used conceptual models and have studied glacial-interglacial periodicity through a shift in climate / ice sheet sensitivity. Legrain et al (2023) were able to capture some characteristics of the MPT by driving the model with only orbital forcing, which were a very interesting comparison to our findings. Willeit et al (2019) used an Earth system model of intermediate complexity, and thus modelled the interactions between ice, climate and the carbon cycle more explicitly.

**Minor comments:**

**Line 8: Provide the approximate timing of the MPT.**

The approximate timing (1.2–0.8 million years ago) was added to line 8.

**Line 17: Here and after you only mention the frequency. Why don't you discuss the amplitude change that should be observed during the MPT?**

We mostly focused on capturing the periodicity and matching the  $\delta^{18}$ O record. However, our lack of substantial sea level amplitude change warrants a broader discussion. We have made a number of changes:

- Whenever we mention "capture glacial-interglacial variability" we instead replace it with "capture glacial-interglacial periodicity" (see line 16), to reflect that we captured specifically the periodicity.
- We added a 1.5-million-year sea level record to figure 4 by Rohling et al. (2021; Science Advances).
- In the discussion, we now suggest possible reasons that could partially explain our lack of amplitude change. These reasons are: (1) our relatively high friction in the Early Pleistocene and (2) the perhaps underestimated CO2 levels in the Yamamoto et al. (2022) record (see lines 343 350).

**Line 19: Here and after: when you are talking about low CO2 levels, are you talking about low glacial or low interglacial CO2levels ? Or both ? Please be more specific to help the reader to correctly understand your findings.**

For each relevant occurrence, we have now specified whether the  $CO_2$  levels reflect interglacial, interstadial or glacial periods (e.g., see line 19). Similarly, we have specified each time whether we refer to constant or prescribed  $CO_2$  levels.

**Line 21: Considering a glacial climate as a "relatively stable climate" could be questioned.**

This can indeed be phrased differently. This statement specifies that this ice sheet configuration (where Laurentide and Cordilleran merge) leads to relatively strong self-sustained growth. A larger (merged) or small ice sheet (separated) state are more sensitive towards increases in insolation or CO2. However, to keep the abstract concise, we have instead removed this statement (See lines 20-24).

**Line 26: I think this is not surprising as the $CO_2$ is part of the forcing index. From which simulation do you derived these results ? Baseline or the comparison between the constant\_ CO2 ?**

These results concern the different constant CO2 concentrations. We made sure that this distinction between baseline and constant\_CO2 simulations are clearer throughout the manuscript. This line was removed (to shorten the abstract) and instead replaced by a statement that constant\_CO2 levels did not yield persistent 100 kyr periodicity (see line 26).

Line 29: Be cautious with this kind of assessments: "The MPT can be explained by", especially in this case: I don't see which simulations can led you to this conclusion, as Yamamoto's record does not register any glacial CO2 decrease and the constant\_CO2 simulation does not make any difference between the interglacial and glacial CO2 decrease. I understand that the abstract is not the appropriate for arguing this, but I think this is a crucial point that deserve more explanation in the discussion.

This sentence was removed, and we made sure to avoid a statement such as "the MPT can be explained by". If low CO2 is maintained during an insolation maximum, it may prevent a termination, thus increase glacial-interglacial periodicity. This is not the same as glacial CO2 decline, and we made sure to not treat it as such in the revised manuscript.

Instead, the abstract now emphasizes that the terminations are triggered if CO2, insolation and ice volume are favourable for a termination (see line 16-24).

Lines 45-52: The three concepts (ice sheet, regolith, and  $CO_2$ ) are nicely explained, but I wouldn't put the first one at the same level as the two others: regolith and  $CO_2$  would be a primary cause of MPT, and the ice sheets are more a secondary response to an initial shift, of what I understand.

The ice sheet threshold regime hypothesis should indeed be viewed not as the prime cause of the MPT, but rather as feedback processes affecting glacial-interglacial periodicity. We now state that the threshold regime hypothesis can "*partially explain the MPT*" (line 45) and also added the following sentence in line 52-54:

"These threshold regimes can therefore act as a precondition that facilitates the MPT, but requires another process (e.g., long term cooling) to prompt a shift in the ice volume rhythm."

We hope that these changes reflect that the ice sheet is a feedback process, but is still very important towards the MPT.

Lines 78-80: Here and after: The most important thing to mention when comparing  $CO_2$  record from ice core to other records is the fact that  $CO_2$  is directly measured in the air trapped in the ice, while other  $CO_2$  records used several transfer function, physical and chemical assumption to go from the initial proxy to the final  $CO_2$  record.

We briefly discussed the difference in uncertainty, but indeed did not mention the challenges of proxy CO2 reconstructions compared to ice cores.

We have now added a few sentences to lines 86-88 to explain this difference between ice core (direct) and proxy records (indirect measurements).

**Line 83: I would mention here that it is the only continuous CO2 record across the MPT, and it is associated with a new and still discussed proxy of CO2 concentrations. The apparently well correlation over the 800-0 ka with ice core record comes from the fact that the amplitude of CO2 concentrations was calibrated on the ice core record itself.**

Indeed, this is also the main reason why we chose the Yamamoto et al (2022) record, as our 1.5 Myr simulations require a transient  $CO_2$  record.

We added the statement "published the first continuous CO2 record covering the past 1.5 million years" and "and calibrated to the 800 kyr ice core record" to the introduction (line 87-90).

**Line 87: Why do you only focus on frequency and not on amplitude? The change of these two parameters is equally crucial in the definition of the MPT.**

During the development of these simulations, we mostly focused on capturing the frequency, rather than capturing amplitude. Additionally, we tuned the model to match the  $\delta^{18}$ O and sea level of the past ~330 kyr, as sea level,  $\delta^{18}$ O and CO2 record at that time are well known.

However, in the results (section 3.1) we only briefly mentioned our too large amplitude. We have now address this in a bit more detail (see 208-216) and compare it to a sea level reconstruction (Rohling et al., 2021; Science Advances). This record was added to figure 4 and figure 8.

There are also possible reasons that could (partially) explain our large Early Pleistocene amplitude. First of all, when applying lower "sediment" friction, the amplitude of Early Pleistocene glacial cycles is reduced by ~10% (the sediment\_change simulation, which were added to the supplementary information). Secondly, boron isotopes and carbon cycle modelling suggest that the Yamamoto et al. (2022) glacial CO2 may be underestimated. If that is the case, this would lead to an underestimation of temperatures, yielding larger ice sheet, and thus partially explaining the large Early Pleistocene amplitude. These two possible reasons are now addressed in the discussion section (see lines 347-351).

Line 107: I understand that everything could not be tested in a single study, and thus the authors choose to keep a constant sediment map. But I would not argue that this approach would help you to solve the answer: does the MPT can be captured without any changes in basal friction? Because the way you will capture the MPT is not a yes/no answer. Reversely, I would say that it would be very interesting to quantitatively described what are the results improvement using a variable sediment mask rather than a constant one (as the approach made by Willeit et al. 2019, science Advances). If there is no significant gain using a variable mask, then you can mention that your model suggests that a variable sediment mask is not relevant to better capture the MPT.

We added the sediment\_change simulation where we apply "homogenous sediment friction" from 1.5 to 0.8 million years ago. Essentially, we replace the friction map with a map that treats the entire domain as if it is filled with sediments. This simulation is similar to the baseline except for a reduction in glacial-interglacial amplitude of ~10%, and a full termination during MIS21 (~865 ka; which is not the case in our baseline). This simulation was added to the supplementary information and is briefly addressed in the results section (see line 212-216). The sediment\_change simulation thus yields a slight improvement.

**Line 167: I appreciate the efforts of the authors to perform an additional baseline\_icecore experiment but I am not sure if this experiment is relevant as the two records are similar over the 800-0 ka period, due to the tunning performed in Yamamoto et al. (2019) on the ice core record.**

This simulation was mostly conducted to show that there is some (but not substantial) difference between the two simulations. But it is true that these provide little to the overall conclusions of the paper.

We have decided to move these experiments to the supplementary information and change the text in section 3.1 accordingly. The simulation was also removed from figure 6.

**Lines 173-175: There is a structure problem in the sentence.**

We fixed this grammatical mistake (line 186).

**Line 183: You go a bit too fast in the description of your experiment: The amplitude of climate cycles are not significantly different before and after the MPT. I think it is a limitation that deserve to be discussed further.**

We made two changes to address this issue:

- We added the sea level reconstruction by Rohling et al. (2021; Science Advances) to figure 4. This also enables us to introduce this limitation in more detail in the results section.
- We addressed two possible reasons behind our large modelled sea level amplitude in the discussion (the low CO2 levels in the proxy record, and our absence of sediment thickness change; see lines 344-351).

**Line 204 and after : The section 3.2 is a combination of results and discussion. Understanding what is deduced from a direct observation is difficult.**

To address this issue, we moved parts of section 3.2 and section 3.3 to the discussion. The discussion section was restructured and rewritten to accommodate these changes. We have addressed a list of changes in major comment (5).

**Line 205: Same remark as for line 107: it is not needed to insist on this point as an argument.**

This line was removed.

**Line 219-220: I do not understand from which result this assumption comes from.**

The "decrease in glacial  $CO_2$  concentrations" should have been "interstadial  $CO_2$  levels" (or "maintaining low  $CO_2$  levels during insolation maximum"). These low  $CO_2$  concentrations can be most clearly seen in figure 4 at 737, 175, 50 kyr ago). We changed this sentence to reflect this (see line 236-230).

**Line 265: Are we still talking about glacial CO2 levels ? Or is it an averaged CO2 concentration level over an entire climate cycle ?**

This statement concerns CO2 levels during insolation maxima.

This entire paragraph was moved to the discussion section and rewritten (see line 352-366). We made sure to be more specific throughout the manuscript whether CO2 levels concerns interglacial, glacial or interstadial.

**Line 276: Why do you chose the present-day insolation value ? Intuitively I would have taken the average insolation value over the past 1.5 Ma.**

We chose present-day for two reasons:

- It is close to the mean caloric summer insolation of the past 1.5 Ma: ~5.79 GJ/yr (0 ka) compared to ~5.84 GJ/yr (average).
- It provides a "realistic" monthly / latitudinal insolation (as this is relevant for the surface melt).

In the revised manuscript, we also added simulations with constant "enhanced" (5 kyr ago), insolation minimum (25 kyr ago), and insolation maximum (10 kyr ago) summer insolation. This should provide a larger range of constant\_insolation values. (See figure 7, and lines 288-304).

**Line 276 and after: This result is interesting, but how would sensitive is it from your chosen insolation value?**

We have now conducted three additional experiments and found that the results are dependent on the insolation value: The constant\_25kyr\_ago\_insolation (insolation minimum), which gave very few terminations. Constant\_10kyr\_ago\_insolation (insolation maximum) generated small ice volumes throughout the simulation.

We also conducted a third simulation: constant\_5kyr\_ago\_insolation, which has enhanced summer insolation compared to present-day. This simulation generates all termination events during the past 1.5 million years, captures the Early Pleistocene ice volume amplitude, but generates long interglacial periods during the Late Pleistocene. Whether the terminations are triggered depends on the interglacial CO2 levels in the Yamamoto et al. (2022) record. The record has low interglacial CO2 levels in the Early Pleistocene (~250 ppm), so it needs relatively strong insolation values to trigger a termination. Only a small range of constant summer insolation strengths can capture all these termination events.

The Yamamoto et al (2022) CO2 record alone can therefore generate the glacial-interglacial periodicity of the past 1.5 million (given a narrow band of summer insolation strengths). We therefore made changes to the results (see lines 288-304), and discussion section (see lines 368-372) to include this new result.

Additionally, we now clearly state in the results (line 293) section that past CO2 levels were affected by insolation. Thereby, the constant\_insolation simulations can still be (indirectly) paced by the orbital cycles.

**Line 288 – 289: Do you perform several simulations (increasing by 10ppm CO2 concentrations) and pick up a posteriori three representative ones, or did you design a priori the experiments with these three specific values for any reason ?**

Prior to the full 1.5-million-year simulations, we conducted the first few 100 thousand years for simulations with 210 - 250 ppm with 10 ppm increment. We selected three that showed strikingly different behaviour and continued these simulations until present-day. Only selecting three therefore provided the most amount of information with the lowest number of simulations.

**Line 304: This assumption clearly could not be part of the result section but should be included in a broader discussion. Also, I would be careful with the use of "explain the MPT".**

This line was removed. We also made sure to avoid statements such as "explain the MPT".

**Line 320: I am not sure to understand how you came to this conclusion.**

While precession is part of the forcing, this signal is completely filtered out in the resulting ice volume. This is caused by non-linear response of the ice sheet towards climate forcing (i.e. the slow build-up and fast self-sustained melt when melt is initiated). This statement was moved to the discussion (line 384-385), and we clarified our explanation (see lines 379-385).

**Discussion section: Some of the results are not contextualized with recent studies that have attempted to model the MPT (e.g. Willeit et al. 2019, Legrain et al. 2023, Verbistky et al. 2018). Some of these articles would reinforce and support the authors' conclusions, or else highlight interesting nuances and diversity in the MPT modeling results.**

These studies are good suggestions. Legrain et al. (2023) has conducted (amongst others) orbital-only forced simulations. They also included ice-volume threshold behaviour to their conceptual model, thereby making this study relevant to compare to our results. Willeit et al. (2019) included active ice sheets, climate and carbon cycle in their simulations, while we use prescribed CO2 changes instead. We added comparison to these studies to the discussion section (see line 356-361 and 374-378).

**Line 332: "CO2 is high enough": would you detailed what you are talking about specifically? It is not obvious that CO2 levels of Early Pleistocene were higher than during**

**late Pleistocene. Especially Yamamoto's record proposes that interglacials CO2levels are higher during late Pleistocene.**

We made sure that statements such as " $CO_2$  is high / low enough" are more specific. In this case, it should have referred to the  $CO_2$  during the insolation maximum. If these are sufficiently high, it can lead to a termination. We have improved our explanation (see lines 339 - 344). Similar issues are also present in other parts of the manuscript (e.g., abstract, discussion), which were resolved.

**Line 333: " $CO_2$ is too low". If this observation comes from one of your simulations, please specify which one (baseline or constant\_ $CO_2$ ). If it comes from the paleodata record, please quote the paper from which you get this information.**

This specific point refers to both the baseline and constant\_CO2. In both cases, if interstadial  $CO_2$  is low, it can prevent additional warming during the insolation maximum, thereby prevent deglaciation and increase the modelled glacial-interglacial periodicity.

This line has been removed and the explanation of our results has been improved. The first paragraph of the discussion (339 – 344) should now more clearly refer to the baseline simulation.

As this is an issue throughout the paper, we made sure that it becomes clearer which conclusion was obtained from the constant\_CO2 or proxy record (Yamamoto et al., 2022).

**Line 334-335: I would be careful about the conclusion coming from these $CO_2$ \_constant simulations. A decreasing $CO_2$ trend throughout the MPT is not similar to successive simulations with constant $CO_2$ levels, but at decreasing values.**

The main conclusions that should be drawn from the constant\_CO2 is that there are fewer terminations when the constant  $CO_2$  levels decrease. This causes increased glacial-interglacial periodicity, though no "true" 100 kyr periodicity is established. We now focus on these points (see lines 372-374), though also be more specific that this concerns constant  $CO_2$  levels, where both interglacial and glacial levels are the same.

**Line 336: I am lost: " $CO_2$ levels have continued to decrease": what are you talking about ? Your simulations ? Which one? Or paleodata records ?**

This refers to a  $CO_2$  concentration that would allow a medium-sized ice sheet to survive during one insolation maximum, but could also yield a collapse of a large ice sheet at the next insolation maximum. The lowest  $CO_2$  level in a glacial cycle (and peak ice volume in reconstructions / our baseline simulation) is often reached just prior to the termination. If the North American ice sheet is large, it is more prone to collapse compared to a "mediumsized' ice sheet. Therefore, despite low  $CO_2$  levels, a large ice sheet may still collapse. As this sentence is confusing, we removed it and instead explain this idea in lines 352-365 in the discussion (while focusing on the baseline simulation).

**Line 364: Please mention the fact that it is an indirect method to reconstruct $CO_2$ concentrations and not a direct measurement.**

We modified the sentence to reflect that  $CO_2$  is measured directly from ice cores, but reconstructed indirectly from leaf wax proxies (see line 392-395). We now explained this in more detail in the introduction (84-88).

Line 368: I strongly disagree with the statement that Yamamoto et al. (2022) and Hönisch et al. (2009) find a decrease in glacial CO2 concentrations. Yamamoto et al. (2022) find constant glacial CO2 concentrations and gradually increasing interglacial CO2concentrations through MPT. Regarding Hönisch et al. (2009), the authors conclude their abstract as following: "atmospheric CO2 did not decrease gradually as would be expected were it to be the driver of the transition.". Nevertheless, it is true that more recent boron isotopes CO2 reconstructions propose a gradual decline of glacial CO2 through the MPT: you would refer to Chalk et al. 2017 (see Fig. 4) rather than Hönisch et al. (2009). The fact that Yamamoto's CO2 record does not evidence any decline of glacial CO2 concentrations is quite problematic for one of the conclusion of this study (a decline of glacial CO2concentrations would have trigger the MPT), as it is the record used as a forcing in the baseline experiment of the authors.

To address these issues, we have made several changes throughout the paper.

First of all, the erroneous statement in line 368 (submitted version) was removed.

Secondly, we have improved our explanation that we obtain longer glacial cycles because low interstadial  $CO_2$  levels are maintained during the insolation maxima. As such, these low  $CO_2$  levels prevent additional warming during the insolation maxima, thus preventing deglaciation (see lines 339-344).

Thirdly, whenever relevant, we more clearly indicate if we refer to interglacial, glacial, or interstadial  $CO_2$  levels. We have also been more specific whether we derive a conclusion from proxy or constant  $CO_2$  levels (e.g., line 26, line 372).

Fourthly, we rewrote and expanded the discussion section to improve the support towards our conclusions, and added additional comparisons to literature (including referring to Chalk et al. (2017) instead of Hönisch et al. (2009) when discussing proxy-based glacial CO2 decline). We now also address that the Yamamoto et al. (2022) record has low CO2 levels compared to recent boron reconstructions and carbon cycle modelling (see lines 344-351).

**Line 369: It is relevant to compare your results to Watanabe, you could do the same for your CO2 results with other modelling studies (e.g. Willeit et al. 2019).**

We have added more comparisons to Willeit et al. 2019, as well as Legrain et al., 2023 and Verbitsky et al., 2018 (line 374-388, line 356-361). Willeit et al. 2019 conducted simulations with an intermediate complexity model that simulates the interactions between ice, climate and carbon cycle. They have also conducted an "orbital forced" only simulations (with freely evolving carbon cycle, but without gradual regolith or  $CO_2$  removal), which yielded 100-kyr periodicity. These are relevant comparisons that have improved the discussion section.

Line 372: I think there is a grammatical problem in the sentence.

This grammatical mistake was fixed (line 401).

**Line 390: I would split the last sentence into two for ease the readability.**

The sentence was split into two (see lines 418-420).

**Figs. 4 and 8: Why not use choose a sea level reconstruction that spanned the last 1.5 Ma ? Here we can not compare your modelled sea level with other reconstructions in the early Pleistocene.**

We have added the sea level reconstruction by Rohling et al., (2021; Science Advances) to both figure 4 and 8, and introduce the differences between our modelled and the reconstructed sea level in section 3.1 (line 210-212).

**Fig. 6: Adding a quantitative x-axis would enhance the readability of the figure.**

We have added a quantitative x-axis, but at the same time also kept the "glacial climate" and "interglacial climate" on the x-axis for easy readability.

Since we conducted additional constant\_insolation simulations for the revised manuscript, we added these to figure 6. We have removed the baseline\_ice\_core experiment, as this became part of the supplementary information.

Additionally, since we added three simulations (and removed only one), we have many additional "onset termination" points. It therefore became more difficult to see which volumes correlate to the largest/least number of terminations. As such, we added a histogram to the figure's background. This histogram indicates the number of "onset of terminations" per ice volume bin.

These changes can be seen in figure 6.

**References:**

Chalk, T. B., Hain, M. P., Foster, G. L., Rohling, E. J., Sexton, P. F., Badger, M. P., ... & Wilson, P. A. (2017). Causes of ice age intensification across the Mid-Pleistocene Transition. Proceedings of the National Academy of Sciences, 114(50), 13114-13119.

Legrain, E., Parrenin, F., & Capron, E. (2023). A gradual change is more likely to have caused the mid-pleistocene transition than an abrupt event. Communications Earth & Environment, 4(1), 90.

Verbitsky, M. Y., Crucifix, M., & Volobuev, D. M. (2018). A theory of Pleistocene glacial rhythmicity. Earth System Dynamics, 9(3), 1025-1043.

Willeit, M., Ganopolski, A., Calov, R., & Brovkin, V. (2019). Mid-Pleistocene transition in

glacial cycles explained by declining CO2 and regolith removal. Science Advances, 5(4), eaav7337.

First of all, we would like to thank the reviewer for their comments on our manuscript. Here we would like to address these concerns. The reviewers' comments are shown in bold; our answers use regular font type instead.

This manuscript studies the causes of the Mid-Pleistocene transition (MPT) with an ice sheet / climate coupled model forced by CO2 variations and orbital variations. The ice sheet model used is IMAU-ICE version 2.1, a vertically integrated model. The transient climate forcing of the ice sheet model is derived using a matrix method by interpolating snapshots of global climate simulations. This method allows us to provide transient climate forcing at a significantly reduced computational time compared to GCM's or intermediate complexity models, albeit without all involved interactions between climate components taken into account. The CO2 forcing used is the one from Yamamoto et al. (2022) based on a leaf-wax indicator. Both sea level and benthic oxygen-18 variations are simulated.

The authors are able to simulate the change of frequency of climate variations during the MPT with this setup. They argue that, because CO2 is the only forcing which has long term variations, it should be the cause of the MPT in the model. They explain that there are 3 regimes of ice sheet variations. Small ice sheet tend to disappear with a relatively small climate forcing. Middle-size ice sheets tend to grow. And large ice sheets are unstable because of several feedbacks that are discussed. They then perform several sensitivity experiments with constant CO2 levels or constant orbital forcing.

Overall, I find the manuscript interesting and well written. The conclusion that a gradual decrease in CO2 is responsible for the MPT coincides with the conclusion of Legrain et al. (Earth. Com. Env., 2023) from a conceptual model. The three regimes of ice sheet variation correspond to what was first proposed by Paillard (1998) and later by Parrenin and Paillard (2003, 2011) with their conceptual models. The model used is quite complex and the work is quite impressive, but the results are presented in a accessible way.

Specific comments:

**I. 207: It should be noted that the model does simulate quite well the change in frequency of sea level variations, but not so well the change in amplitude.**

This issue was only briefly addresses in section 3.1. Our main focus was to capture the frequency change during the MPT. Therefore, we have made several changes to the manuscript to address the lack of amplitude change:

- In the abstract we replaced the line *"capture glacial-interglacial variability"* with *"capture glacial-interglacial periodicity"* (line 16). As we simulated the frequency change, but did not obtain substantial amplitude change.

Similarly, in line 92 we changed the sentence to: "Our main goal is to explore if we can simulate the frequency change during the MPT".

We have now added a comparison to a sea level reconstruction by Rohling et al., (2021; Science Advances) to figure 4 (time-series of the baseline simulation). It covers the entire 1.5-million-year simulation.

- In the discussion section, we now propose two reasons that could partially explain our too-large amplitude in the Early Pleistocene (see lines 344-351):

Firstly, we have now completed a simulation with low (sediment) friction during the Early Pleistocene, which shows a ~10% reduction in amplitude. This run was added to the supplementary information, and briefly introduced in the results section (line 194).

Secondly, the CO2 concentrations in the Yamamoto et al., (2022) reconstruction are low compared to boron isotope and carbon cycle modelling studies (e.g., Chalk et al., 2017; Willeit et al., 2019). If the Yamamoto et al. (2022) CO2 levels are indeed underestimated in the Early Pleistocene, this could partially explain our large ice volume during that period.

**I. 209: There are also long term amplitude modulations in the orbital forcing. Legrain et al. (2023) have a simulation of the MPT with only orbital forcing, which is less realistic than the simulation with a long-term forcing, but which still contain some sort of frequency change.**

Legrain et al. (2023) has conducted conceptual model simulations and obtained a change in glacial-cycle frequency in the ORB (only orbital forcing) experiment. This shows that they were able to capture some characteristics of the MPT, without any additional drivers.

The claim "Since the orbital cycles cannot explain the MPT" is therefore too strong. We replaced the sentence with "Since the orbital cycles alone cannot explain all characteristics of the MPT" (see line 225-227).

We now also discuss Legrain et al. (2023) in the discussion section (line 374-375 and 359-361).

**l. 252: "50 m.s.l.e."**

This number refers to the "gap" where for a certain range of climate forcing and ice volumes we obtained few deglaciations. To prevent confusion, we have replaced the line "*between 40 and 55 m.s.l.e.*" (meter sea level equivalent) with "*around 50 m.s.l.e.*" (see line 266). Additionally, we have added histograms in the background of figure 6a, which should make this lack of terminations at 50 m.s.l.e. clearer.

**I. 372: "Simulating 1.5 million years requires..."**

This grammar mistake has been fixed (see line 401).

**I. 381: it is a bit surprising to have a discussion section but no conclusion.**

We have added a brief conclusion section (see lines 421 - 444).

---

## Referee Report (RR1)

Review of CP-2024-57

**CO2 and summer insolation as drivers for the Mid-Pleistocene transition**

by Meike D. W. Scherrenberg et al.

**General Evaluation:**

The study provides a well-organized review of previous research, systematically summarizing the relationships between insolation, CO2, regolith deposits, and the periodicity of glacial–interglacial cycles. The motivation behind this study is clearly articulated: to investigate the causes of the Mid-Pleistocene Transition (MPT) by conducting baseline experiments based on reconstructions and examining idealized responses to insolation and CO2, thereby assessing their respective contributions.

The classification of ice sheets into three distinct stages, considering their size, shape, and susceptibility to melting (i.e., the likelihood of termination), is particularly insightful. The study suggests that a threshold exists whereby large ice sheets are more prone to abrupt changes even under a glacial climate, which is a compelling finding.

By integrating baseline experiments with idealized insolation and CO2 experiments, the study presents key interpretations: (1) even under low CO2 conditions, high insolation can still induce ice sheet melting and lead to deglaciation, (2) without sufficiently low CO2 levels, prolonged glacial periods cannot be sustained, and (3) the variations in glacial–interglacial cycles observed since the MPT can be explained by changes in CO2 levels.

Furthermore, the study discusses the regolith hypothesis, which posits that variations in bedrock friction influence the ease of ice sheet melting. However, the methodology for idealizing the friction coefficient changes remains somewhat unclear. Specifically, it is not entirely evident what real-world conditions this experimental setup aims to replicate, what assumptions underlie it, and how it contributes to the broader discussion of the regolith hypothesis. Clarifying these aspects would enhance the study's interpretation and its implications for understanding the role of basal friction in glacial–interglacial dynamics.

Regarding the methods section, it would be helpful if the basic settings of the model were explained in more detail rather than simply referencing citations. For example, in the following sentence, it is unclear what is meant by "*do not vary spatially within a model domain.*" Providing a clearer and more detailed explanation of this point would improve the reader's understanding: "*Ocean temperatures are based on de Boer et al.* (2013), and while they evolve over time, they do not vary spatially within a model domain."

Finally, as noted by the other reviewer, the conclusion lacks clarity regarding the precise stance of the study. It remains unclear what the key takeaway is in relation to previous research. Specifically, to what extent does this study introduce novel insights? Is it questioning the conventional view that variations in orbital parameters and the resulting insolation changes are the primary drivers of glacial–interglacial cycles? Is the key finding that CO2 and insolation become important at different phases of the cycles? Or is the main argument that sufficiently high interglacial CO2 levels are crucial for the observed lengthening of glacial cycles?

Currently, the significance of the study is not conveyed as a clear and compelling message. Strengthening this aspect would enhance the impact of the paper, ensuring that readers fully grasp its contribution to the broader understanding of glacial–interglacial dynamics.

In any case, the manuscript has been well revised in response to the previous reviewers' comments, and I believe it has reached a level worthy of publication.

**Minor Concerns:**

L130: "all data necessary for our simulations" What kind of data do you use here?

L185: How do the authors explain the fact that the omission of the Antarctic ice sheet in the baseline experiment has led to a 20% overestimation of sea level change?

L213: What is the justification for the experimental setup of "sediment"? A more detailed discussion on the extent to which these results support or challenge the regolith hypothesis would enhance the clarity and impact of the study's conclusions.

Fig2:What is the meaning of forcing index?

---

## Author Response (AR2)

First of all, we would like to thank the reviewer for their comment on our revised comments. The reviewers' comments are shown in bold, while our answers are shown in regular font-type.

I acknowledge the authors for their work on this revised version of the manuscript. The readability of the manuscript, particularly in the results and discussion sections, has been significantly improved, and the content is now more focused and specific. Additionally, the authors conducted further simulations that clarify some of the key results. They have also addressed the role of CO2 during the MPT, which was not clearly defined in the initial version of the manuscript. In my opinion, the paper is almost ready for publication.

My only remaining minor concern is the lack of explanation regarding the authors decision to focus solely on "frequency changes" in the MPT. While I appreciate the challenges models face in accurately representing both the amplitude and frequency of the MPT, I would advise caution in treating these two parameters as entirely independent. Including a brief statement acknowledging that all results and interpretations related to MPT frequency are based on simulations that do not capture its amplitude would provide important context and ensure fairness in the interpretation of the findings.

We added a statement to the discussion section that our results may be influenced by the lack of substantial amplitude change. We put this limitation in the context of one of our other results: the threshold for ice sheet collapse depends on the ice volume and climate forcing (See line 357-359).

In conclusion, I congratulate the authors on the large amount of work produced in their study, which provides valuable contribution to the understanding of the mechanisms at the origin of the MPT.

We would like to thank the reviewer for their comments on our revised comments. The reviewers' comments are shown in bold; our answers are shown in regular font-type.

Clarification of the Regolith Hypothesis: The methodology for idealizing changes in basal friction is somewhat unclear. It would be beneficial to explicitly state how the experimental setup reflects real-world conditions and what assumptions underlie it.

We added a few sentences at line 217 to clarify the method used for this reduced friction simulation. We emphasized that our method is a simplistic test for decreased friction during the Early Pleistocene.

Detailed Explanation of Model Settings: The methods section relies heavily on citations without sufficient detail on basic model settings. For example, the statement that ocean temperatures "do not vary spatially within a model domain" needs further clarification to aid reader comprehension.

We added a few elaborations in the method's section:

We specified the basal sliding method and added a reference in line 111.

We explained that the sub-grid friction scheme could capture Marine / Proglacial ice sheet instability (line 120).

We more clearly stated that the ocean temperatures are homogenous and are interpolated with respect to  $CO_2$  and insolation (see line 124).

In lines 127, we specified that the snow-rain partitioning scheme is temperature-based, and the amount of refreezing is limited to the available liquid water, temperature and firn depth.

Stronger Conclusion and Contribution: The key takeaway of the study in relation to previous research remains somewhat ambiguous. Clarifying whether the study primarily challenges the conventional role of orbital forcing, emphasizes phase-dependent importance of  $CO_2$  and insolation, or highlights interglacial  $CO_2$  thresholds as a key driver of glacial cycle lengthening would significantly improve the impact of the conclusions.

We have decided to rewrite the conclusion section to clarify our key takeaways. To improve clarity, we have summarized our conclusions using bullet-points. Additionally, the conclusions section now mainly focuses on the threshold behavior between CO2, insolation and ice sheet volume. In the introduction, we also added a brief statement that our study is the first one that conducted ice-sheet model simulations using the Yamamoto et al. (2022) dataset (see line 97).